# LINKLESS LINK PREDICTION VIA RELATIONAL DISTILLATION

## ABSTRACT

Graph Neural Networks (GNNs) have been widely used on graph data and have shown exceptional performance in the task of link prediction. Despite their effectiveness, GNNs often suffer from high latency due to non-trivial neighborhood data dependency in practical deployments. To address this issue, researchers have proposed methods based on knowledge distillation (KD) to transfer the knowledge from teacher GNNs to student MLPs, which are known to be efficient even with industrial scale data, and have shown promising results on node classification. Nonetheless, using KD to accelerate link prediction is still unexplored. In this work, we start with exploring two direct analogs of traditional KD for link prediction, i.e., predicted logit-based matching and node representation-based matching. Upon observing direct KD analogs do not perform well for link prediction, we propose a relational KD framework, *Linkless Link Prediction* (LLP). Unlike simple KD methods that match independent link logits or node representations, LLP distills *relational* knowledge that is centered around each (anchor) node to the student MLP. Specifically, we propose two matching strategies that complement each other: rank-based matching and distribution-based matching. Extensive experiments demonstrate that LLP boosts the link prediction performance of MLPs with significant margins, and even outperforms the teacher GNNs on 6 out of 9 benchmarks. LLP also achieves a $776.37\times$ speedup in link prediction inference compared to GNNs on the large scale `OGB-Citation2` dataset.

## 1 INTRODUCTION

Graph neural networks (GNNs) have been widely used for machine learning on graph-structured data (Kipf & Welling, 2016a; Hamilton et al., 2017). They have shown significant performance in various applications, such as node classification (Veličković et al., 2017; Chen et al., 2020), graph classification (Zhang et al., 2018; Ying et al., 2018b), graph generation (You et al., 2018; Shiao & Papalexakis, 2021), and link prediction (Zhang & Chen, 2018).

Of these, link prediction is a notably critical problem in the graph machine learning community, which aims to predict the likelihood of any two nodes forming a link. It has broad practical applications such as knowledge graph completion (Schlichtkrull et al., 2018; Nathani et al., 2019; Vashishth et al., 2020), friend recommendation on social platforms (Sankar et al., 2021; Tang et al., 2022; Fan et al., 2022) and item recommendation for users on service and commerce platforms (Koren et al., 2009; Ying et al., 2018a; He et al., 2020). With the rising popularity of GNNs, state-of-the-art link prediction methods adopt encoder-decoder style models, where encoders are GNNs, and decoders are applied directly on pairs of node representations learned by the GNNs (Kipf & Welling, 2016b; Zhang & Chen, 2018; Cai & Ji, 2020; Zhao et al., 2022).

The success of GNNs is typically attributed to the explicit use of contextual information from nodes' surrounding neighborhoods (Zhang et al., 2020e). However, this induces a heavy reliance on neighborhood fetching and aggregation schemes, which can lead to high time cost in training and inference compared to tabular models, such as multi-layer perceptrons (MLPs), especially owing to neighbor explosion (Zhang et al., 2020b; Jia et al., 2020; Zhang et al., 2021b; Zeng et al., 2019). Compared to GNNs, MLPs do not require any graph topology information, making them more suitable for new or isolated nodes (e.g., for cold-start settings), but usually resulting in worse general task performance as encoders, which we also empirically validate Section 4. Nonetheless, having

no graph dependency makes the training and inference time for MLPs negligible when comparing with those of GNNs. Thus, in industrial-scale applications where fast real-time inference is required, MLPs are still a leading option (Zhang et al., 2021b; Covington et al., 2016; Gholami et al., 2021).

Given these speed-performance tradeoffs, several recent works propose to transfer the learned knowledge from GNNs to MLP using knowledge distillation (KD) techniques (Hinton et al., 2015; Zhang et al., 2021b; Zheng et al., 2021; Hu et al., 2021), to take advantage of both GNN's performance benefits and MLP's speed benefits. Specifically, in this way, the student MLP can potentially obtain the graph-context knowledge transferred from the GNN teacher via KD to not only perform better in practice, but also enjoy model latency benefits compared to GNNs, e.g. in production inference settings. However, these works focus on node- or graph-level tasks. Given that KD on link prediction tasks have not been explored, and the massive scope of recommendation systems contexts that are posed as link prediction problems, our work aims to bridge a critical gap. Specifically, we ask:

*Can we effectively distill link prediction-relevant knowledge from GNNs to MLPs?*

In this work, we focus on exploring, building upon, and proposing cross-model (GNN to MLP) distillation techniques for link prediction settings. We start with exploring two direct KD methods of aligning student and teacher: (i) logit-based matching of predicted link existence probabilities (Hinton et al., 2015), and (ii) representation-based matching of the generated latent node representations (Gou et al., 2021). However, empirically we observe that neither the logit-based matching nor the representation-based matching are powerful enough to distill sufficient knowledge for the student model to perform well on link prediction tasks. We hypothesize that the reason of these two KD approaches not performing well is that link prediction, unlike node classification, heavily relies on *relational* graph topological information (Martínez et al., 2016; Zhang & Chen, 2018; Yun et al., 2021; Zhao et al., 2022), which is not well-captured by direct methods.

To address this issue, we propose a relational KD framework, namely LLP: our key intuition is that instead of focusing on matching individual node pairs or node representations, we focus on matching the relationships between each (anchor) node with respect to other (context) nodes in the graph. Given the relational knowledge centered at the anchor node, i.e., the teacher model's predicted link existence probabilities between the anchor node and each context node, LLP distills it to the student model via two matching methods: (i) rank-based matching, and (ii) distribution-based matching. More specifically, rank-based matching equips the student model with a ranking loss over the relative ranks of all context nodes w.r.t the anchor node, preserving crucial ranking information that are directly relevant to downstream link prediction use-cases, e.g. user-contextual friend recommendation (Sankar et al., 2021; Tang et al., 2022) or item recommendation (Ying et al., 2018a; He et al., 2020). On the other hand, distribution-based matching equips the student model with the link probability distribution over context nodes, conditioned on the anchor node. Importantly, distribution-based matching is complementary to rank-based matching, as it provides auxiliary information about the relative values of the probabilities and magnitudes of differences.

To comprehensively evaluate the effectiveness of our proposed LLP, we conduct experiments on 9 public benchmarks. In addition to the standard transductive setting for graph tasks, we also design a more realistic setting that mimics realistic (on-line) use-cases for link prediction, which we call the production setting. LLP consistently outperforms stand-alone MLPs by 17.13 points on average under the transductive setting and 12.01 points under the production setting on all the datasets, and matches or outperforms teacher GNNs on 6/9 datasets under the transductive setting. Promisingly, for cold-start nodes, LLP outperforms teacher GNNs and stand-alone MLPs by **25.29** and **9.42** Hits@20 on average, respectively. Finally, LLP infers drastically faster than GNNs, e.g. **776.37**$\times$ faster on the large-scale `OGB-Citation2` dataset.

## 2 RELATED WORK AND PRELIMINARIES

We briefly discuss related work and preliminaries relevant to contextualizing our methods and contributions. Due to space limit, we defer more related work to Appendix A.

**Notation.** Let $G = (\mathcal{V}, \mathcal{E})$ denote an undirected graph, where $\mathcal{V}$ denotes the set of $N$ nodes and $\mathcal{E} \subseteq \mathcal{V} \times \mathcal{V}$ denotes the set of observed links. $\mathbf{A} \in \{0, 1\}^{N \times N}$ denotes the adjacency matrix, where $\mathbf{A}_{i,j} = 1$ if exists an edge $e_{i,j}$ in $\mathcal{E}$ and 0 otherwise. Let the matrix of node features be denoted by $\mathbf{X} \in \mathbb{R}^{N \times F}$, where each row $\boldsymbol{x}_i$ is the $F$-dim raw feature vector of node $i$. Given both $\mathcal{E}$ and $\mathbf{A}$ have

the validation and test links masked off for link prediction, we use $a_{i,j}$ (different from $\boldsymbol{A}_{i,j}$) to denote the true label of link existence of nodes $i$ and $j$, which may or may not be visible during training depending on the setting. We use $\mathcal{E}^- = (\mathcal{V} \times \mathcal{V}) \setminus \mathcal{E}$ to denote the no-edge node pairs that are used as negative samples during model training. We denote node representations by $\mathbf{H} \in \mathbb{R}^{N \times D}$, where $D$ is the hidden dimension. In KD context with multiple models, we use $\boldsymbol{h}_i$ and $\hat{\boldsymbol{h}}_i$ to denote node $i$'s representations learned by the teacher and student models, respectively. Similarly, we use $y_{i,j}$ and $\hat{y}_{i,j}$ to denote the predictions for $a_{i,j}$ by the teacher and the student models, respectively.

**Link Prediction with GNNs.** In this work, we take the commonly used encoder-decoder framework for the link prediction task (Kipf & Welling, 2016b; Berg et al., 2017; Schlichtkrull et al., 2018; Ying et al., 2018a; Davidson et al., 2018; Zhu et al., 2021; Yun et al., 2021; Zhao et al., 2022), where the GNN-based encoder learns node representations and the decoder predicts link existence probabilities. Most GNNs operate under the message-passing framework, where the model iteratively updates each node $i$'s representation $\boldsymbol{h}_i$ by fetching its neighbors' information. That is, the node's representation in the $l$-th layer is learned with an aggregation operation and an update operation:

$$\boldsymbol{h}_i^l = \text{UPDATE}_l\big(\boldsymbol{h}_i^{l-1}, \text{AGGREGATE}_l(\{\boldsymbol{h}_j^{l-1} | e_{i,j} \in \mathcal{E}\})\big), \tag{1}$$

where AGGREGATE combines or pools local neighbor features, UPDATE is a learnable transformation, and $\boldsymbol{h}_i^0 = \boldsymbol{x}_i$. The link prediction decoder takes the node representations from the last layer, i.e., $\boldsymbol{h}_i$ for $i \in \mathcal{V}$, to predict the probability of a link between a node pair $i$ and $j$:

$$y_{i,j} = \sigma(\text{DECODER}(\boldsymbol{h}_i, \boldsymbol{h}_j)), \tag{2}$$

where $\sigma$ denotes a Sigmoid operation. In this work, following most state-of-the-art link prediction literature (Zhang et al., 2021a; Tsitsulin et al., 2018; Zhao et al., 2022; Wang et al., 2021), we take the Hadamard product followed by a MLP as the link prediction DECODER for all methods.

**Knowledge Distillation for GNNs.** Knowledge distillation (KD) (Hinton et al., 2015) aims to transfer the knowledge from a high-capacity and highly accurate teacher model to a light-weight student model, and is typically employed in resource-constrained settings. KD methods have shown great promise in significantly reducing model complexity, while sometimes barely (or not) sacrificing task performance (Furlanello et al., 2018; Park et al., 2019). As GNNs are known to be slow due to neighbor aggregation induced by data dependency, graph-based KD methods (Zhang et al., 2021b; Zheng et al., 2021) usually distill GNNs onto MLPs, which are commonly used in large-scale industrial applications due to their significantly improved efficiency and scalability. For example, Zheng et al. (2021) proposed a KD-based framework to rediscover the missing graph structure information for MLPs, which improves the models' generalization of node classification task on cold-start nodes. Existing KD methods on graph data mainly focus on node-level (Zheng et al., 2021; Zhang et al., 2021b; Tian et al., 2022) and graph-level tasks (Ma & Mei, 2019; Zhang et al., 2020c; Deng & Zhang, 2021; Joshi et al., 2021), leaving KD for link prediction yet unexplored. Our work focuses on distilling link prediction-relevant information from the GNN teacher to an MLP student, and investigates various KD strategies to align student and teacher predictions. Specifically, denoting the node representations for nodes $i$ and $j$ learned by the student MLP as $\hat{\boldsymbol{h}}_i$ and $\hat{\boldsymbol{h}}_j$, the link existence prediction by the student model can then be written as $\hat{y}_{i,j} = \sigma(\text{DECODER}(\hat{\boldsymbol{h}}_i, \hat{\boldsymbol{h}}_j))$.

## 3  CROSS-MODEL KNOWLEDGE DISTILLATION FOR LINK PREDICTION

In this section, we propose and discuss several approaches to distill knowledge from a teacher GNN to a student MLP in a cross-model fashion, for the purpose of link prediction. In all cases, we aim to supervise the student MLP with artifacts produced by the GNN teacher, in addition to any available training labels ($a_{i,j}$ w.l.o.g.) about link existence. We start by adapting two *direct* knowledge distillation (KD) methods: logit-matching and representation-matching, on link prediction tasks; we call these methods direct because they involve directly matching sample-wise predictions between teacher and student. Next, we motivate and introduce our proposed *relational* KD framework, LLP, with two matching strategies to distill additional topology-related structural information to the student. We call these methods relational because they call for preservation of relationships across samples between teacher and student (Park et al., 2019). We next discuss our proposals in more detail, which are also summarized in Figure 1.

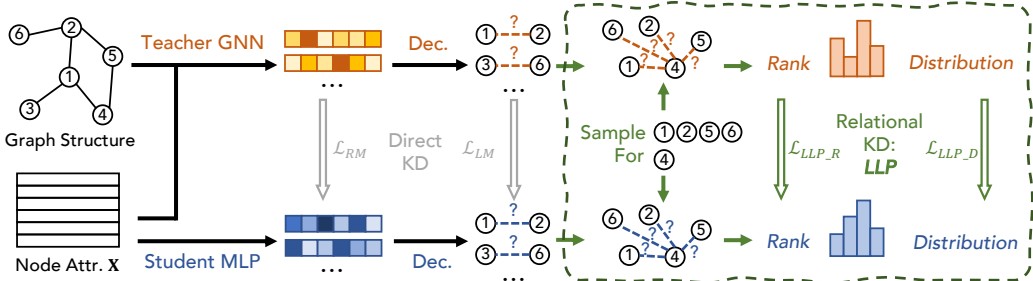

Figure 1: We explore KD methods for link prediction, which distill knowledge from a Teacher GNN to a Student MLP encoder, each with their own decoder. We start by exploring direct KD methods: representation-matching and logit-matching. Upon observing their drawbacks of not being able to distill relational information, we further propose a relational KD framework: LLP, which equips the student model with knowledge of each (anchor) node's relationships with other (context) nodes, via our proposed rank-based matching and distribution-based matching objectives.

## 3.1 DIRECT METHODS

**Logit-matching** is one straightforward strategy to distill knowledge from the teacher to the student, where it directly aims to teach the student to generalize as the teacher does on the downstream task. It (Hinton et al., 2015) was originally proposed some time ago, but it is still one of the most widely used KD methods in various tasks (Furlanello et al., 2018; Yang et al., 2020b; Yan et al., 2020). Several works (Phuong & Lampert; Ji & Zhu, 2020) theoretically analyzed its effectiveness. Moreover, it had also been proved to be effective for knowledge transfer on graph data (Yan et al., 2020; Yang et al., 2021; Zhang et al., 2021b) in recent years. For example, Zhang et al. (2021b) the soft logits generated by the teacher GNNs to help supervise the student MLP and achieved strong performance on node classification tasks. In a similar vein, we generate the soft score $y_{i,j}$ for the node pair or candidate edge $(i, j)$ with the teacher GNN model, and train the student to match its prediction $\hat{y}_{i,j}$ on this target:

$$\mathcal{L}_{LM} = \sum_{(i,j) \in \mathcal{E} \cup \mathcal{E}^-} \lambda \mathcal{L}_{sup}(\hat{y}_{i,j}, a_{i,j}) + (1 - \lambda) \mathcal{L}_{match}(\hat{y}_{i,j}, y_{i,j}), \qquad (3)$$

where $\mathcal{L}_{sup}$ is the supervised link prediction loss (e.g., binary cross entropy) that directly trains the student model, $\mathcal{L}_{match}$ is the loss for matching the student's prediction with the teacher's prediction, and $\lambda$ is a hyper-parameter that mediates the importance of the ground-truth labels and logit-matching signals. Note that multiple implementation choices exist for $\mathcal{L}_{match}$. For example, mean-squared error (MSE), Kullback-Leibler (KL) divergence, or cosine similarity. In the experiments, we opt for the empirical best choice for fair comparison across methods.

**Representation-matching** is another direct distillation method in which we aim to align the student's learned latent node embedding space with the teacher's. As this KD training signal only optimizes the encoder part of the student model, it must be used in conjunction with $\mathcal{L}_{sup}$ so that the student decoder receives a gradient and can also be optimized:

$$\mathcal{L}_{RM} = \sum_{(i,j) \in \mathcal{E} \cup \mathcal{E}^-} \lambda \mathcal{L}_{sup}(\hat{y}_{i,j}, a_{i,j}) + (1 - \lambda) \sum_{i \in \mathcal{V}} \mathcal{L}_{match}(\hat{\boldsymbol{h}}_i, \boldsymbol{h}_i). \qquad (4)$$

Unlike logit-matching, representation-matching involves directly aligning node-level artifacts, which is similar to object representation matching in computer vision (Romero et al., 2014; Kim et al., 2018; Wang et al., 2020b; Chen et al., 2021a).

## 3.2 LINK PREDICTION WITH RELATIONAL DISTILLATION

**Motivation.** The above direct methods ask the student model to directly match node-level or link-level artifacts. However, one might ask: are matching these sufficient for link prediction tasks? This is especially relevant considering that most link prediction applications involve tasks where

ranking target nodes with respect to a source, or anchor node, is the task of interest, i.e. ranking relevant candidate users or items with respect to a seed user (Huang et al., 2005; Trouillon et al., 2016). In other words, these contexts involve reasoning over multiple *relations* or link-level samples simultaneously, suggesting that matching across these relations could be more aligned with the target link prediction task, compared to the direct node-level or link-level methods.

Furthermore, several works (Zhang & Chen, 2018; Yun et al., 2021) suggest that graph structure information is critically important for link prediction tasks. For example, heuristic link prediction methods commonly show competitive performance compared to GNNs (Zhang & Chen, 2018) and have long-served as a cornerstone for accurate link prediction even prior to neural graph methods (Martínez et al., 2016). Most heuristic methods measure the score of the target node pairs only based on the graph structure information (Barabási & Albert, 1999; Brin & Page, 2012), such as common neighbors and shortest path. In addition, several recent works (Zhang & Chen, 2017; 2018; Li et al., 2020; Zhao et al., 2022) also show that enclosing topology information such as local subgraph, distances with anchor nodes, or augmented links can largely improve GNNs' performance on link-level tasks. Observing that most successful methods in link prediction involve using relational information other than just the two nodes in question, we also adopt this intuition in the distillation context, and propose our relational KD for link prediction. We elaborate next.

### 3.3 PROPOSED FRAMEWORK: LINKLESS LINK PREDICTION

In accordance with our intuition regarding preservation of relational knowledge, we propose a novel relational distillation framework, called *Linkless Link Prediction*, or LLP. Instead of focusing on matching individual node pair scores or node representations, LLP focuses on distilling knowledge about the relationships of each node to other nodes in the graph; we call the former node an *anchor* node, and the latter nodes *context* nodes. For each node in the graph, when it serves as the anchor node, we aim to equip the student MLP model with knowledge of its relationships with a set of context nodes. In practice, each node can serve as both an anchor node, as well as a context node (for other anchor nodes).

Let $v$ denote the anchor node and $\mathcal{C}_v$ denote the corresponding set of context nodes of $v$. We denote the teacher model's predicted probabilities of $v$ and each node in $\mathcal{C}_v$ as $\mathcal{Y}_v = \{y_{v,i} | i \in \mathcal{C}_v\}$. Similarly, we denote the student model's predictions on those as $\hat{\mathcal{Y}}_v = \{\hat{y}_{v,i} | i \in \mathcal{C}_v\}$. To effectively distill the relational knowledge from $\mathcal{Y}_v$ to $\hat{\mathcal{Y}}_v$, we proposed two relational matching objectives to train LLP: *rank-based matching* and *distribution-based matching*, which we introduce next.

**Rank-based Matching.** As aforementioned in Section 3.2, link prediction is often considered a ranking task, requiring the model to rank relevant candidates w.r.t. a seed node, e.g. in a user-item graph setting, the predictor must rank over a set of candidate items from the perspective of a user. Thus, we reason that unlike matching individual and independent logits, matching the ranking induced by the teacher can more straightforwardly teach the student relational knowledge about context nodes w.r.t. the anchor node, e.g. for a specific user, item $A$ should be ranked higher than item $C$, which should be ranked higher than item $B$. To adopt this rank-based intuition into a training objective, we adopt a modified margin-based ranking loss that trains the student with the rank of the logits from the teacher GNN. Specifically, we enumerate all pairs of predicted probabilities in $\hat{\mathcal{Y}}_v$ and supervise it with the corresponding pairs in $\mathcal{Y}_v$. That is,

$$\mathcal{L}_{\text{LLP\_R}} = \sum_{v \in \mathcal{V}} \sum_{\{\hat{y}_{v,i}, \hat{y}_{v,j}\} \in \hat{\mathcal{Y}}_v} \max(0, -r \cdot (\hat{y}_{v,i} - \hat{y}_{v,j}) + \delta), \tag{5}$$

$$\text{where} \quad r = \begin{cases} 1, & \text{if } y_{v,i} - y_{v,j} > \delta; \\ -1, & \text{if } y_{v,i} - y_{v,j} < -\delta; \\ 0, & \text{otherwise,} \end{cases}$$

where $\delta$ is the margin hyper-parameter, which is usually a very small value (e.g. 0.05). Note that the above loss differs from the conventional margin-based ranking loss, because it has a condition for $r = 0$ (inducing constant loss) on the logits pairs that the teacher GNN gives similar probabilities, i.e., $|y_{v,i} - y_{v,j}| < \delta$. This design effectively prevents the student model from trying to differentiate minuscule differences in probabilities which the teacher may produce owing to noise; without this condition, the loss would pass binary information regardless of how small the difference is. We also empirically show the necessity of this design in Table 9 in Appendix D.

**Distribution-based Matching.** While the rank-based matching can effectively teach the student model relational rank information, we observe that it does not fully make use of the value information from $\mathcal{Y}_v$, e.g. for a specific user, item $A$ should be ranked *much* higher than item $C$, which should only be ranked *marginally* higher than item $B$. Although the logit-matching introduced in Section 3.1 might seem suitable here, we observe that its link-level matching strategy only facilitates matching information on scattered node pairs, rather than focusing on the relationships conditioned on an anchor node – empirically, we also find that it has limited effectiveness. Therefore, to enable relational value-based matching centered on the anchor nodes, we further propose a distribution-based matching scheme which utilizes the KL divergence between the teacher predictions $\mathcal{Y}_v$ and student predictions $\hat{\mathcal{Y}}_v$, centered on each anchor node $v$. Specifically, we define it as

$$\mathcal{L}_{\mathsf{LLP\_D}} = \sum_{v \in \mathcal{V}} \sum_{i \in \mathcal{C}_v} \frac{\exp(y_{v,i}/\tau)}{\sum_{j \in \mathcal{C}_v} \exp(y_{v,j}/\tau)} \log \left( \frac{\exp(\hat{y}_{v,i}/\tau)}{\sum_{j \in \mathcal{C}_v} \exp(\hat{y}_{v,j}/\tau)} \right), \qquad (6)$$

where $\tau$ is a temperature hyper-parameter which controls the softness of the softmaxed distribution. By also asking the student to match relative values within the probability distribution over context nodes conditioned on each anchor node, the distribution-based matching scheme complements rank-based matching by providing auxiliary information about the magnitudes of differences.

**Practical Implementation of LLP.** In practical implementation, given the large number of nodes in the graph, it is infeasible for LLP to use all other nodes as the set of context nodes, especially for the rank-based matching which enumerates pairs of probabilities in $\hat{\mathcal{Y}}_v$. Hence, we opt for simplicity and adopt two straightforward sampling strategies for the constructing $\mathcal{C}_v$ for each anchor node $v$ to limit its size. First, to preserve the local structure around $v$, we sample $p$ nearby nodes via fixed-length random walks, denoted as $\mathcal{C}_v^N$. On the other hand, we also randomly sample $q$ nodes from $G$ (which are likely to be far-away from $v$) to form $\mathcal{C}_v^R$, which additionally preserves the global structure w.r.t. $v$ in the graph. $p$ and $q$ are hyper-parameters. Finally, we make $\mathcal{C}_v$ as the union of the nearby samples and random samples, i.e., $\mathcal{C}_v = \mathcal{C}_v^N \cup \mathcal{C}_v^R$. We note that LLP can easily adopt more sophisticated sampling strategies for further improvements; we leave this to future exploration.

While training LLP, we jointly optimize both the rank-based and distribution-based matching losses in addition to the ground-truth label loss. Therefore, the overall training loss which LLP adopts for the student is

$$\mathcal{L} = \alpha \cdot \mathcal{L}_{sup} + \beta \cdot \mathcal{L}_{\mathsf{LLP\_R}} + \gamma \cdot \mathcal{L}_{\mathsf{LLP\_D}} \qquad (7)$$

where $\alpha$, $\beta$, and $\gamma$ are hyper-parameters which mediate the strengths of each loss term.

# 4 EXPERIMENTS

## 4.1 EXPERIMENTAL SETUP

**Datasets.** We conduct the experiments using 9 benchmark datasets, including 7 commonly used link prediction benchmarks (Cora, Citeseer, Pubmed, Computers, Photos, CS, and Physics) and 2 larger-scale link prediction benchmarks from OGB (Hu et al., 2020) (OGB-Collab and OGB-Citation2). Further details and statistics of the datasets are summarized in Appendix B.

**Evaluation Settings and Metrics.** To comprehensively evaluate our proposed LLP and baseline methods on the link prediction tasks, we conduct experiments on both transductive and production settings. For the transductive setting, all the nodes in the graph can be observed for train/validation/test sets. Following previous works (Zhang & Chen, 2018; Chami et al., 2019; Cai et al., 2021) we randomly sample 5%/15% of the links with the same number of no-edge node pairs from the graph as the validation/test sets on the non-OGB datasets. And the validation/test links are masked off from the training graph. For the OGB datasets, we follow their official train/validation/test splits (Wang et al., 2020a; Mikolov et al., 2013). In addition to transductive setting, we also design a more realistic setting that mimics practical link prediction use-cases, which we call the production setting. In the production setting, new nodes would appear in the test set, while training and validation sets only observe previously existing nodes. Thus, this setting entails three categories of node pairs (edges or no-edges) in the test set: existing – existing, existing – new, and new – new, where the first category is similar to the test edges in the transductive setting, and the latter two categories together are similar to the inductive setting used in a few recent works (Bojchevski & Günnemann, 2017; Hao et al., 2020;

Table 1: Link prediction performance under transductive setting. For `OGB-Collab` and `OGB-Citation2`, we report Hits@50 and MRR respectively. For other datasets, we report Hits@20. Best and second best performances are marked with bold and underline, respectively. $\Delta_{DirectKD}$, $\Delta_{MLP}$, and $\Delta_{GNN}$ represent differences between LLP and these methods.

| | GNN | MLP | $\mathcal{L}_{LM}$ | $\mathcal{L}_{RM}$ | LLP | $\Delta_{DirectKD}$ | $\Delta_{MLP}$ | $\Delta_{GNN}$ |
|---|---|---|---|---|---|---|---|---|
| Cora | $74.38_{\pm1.54}$ | $\underline{78.06}_{\pm1.50}$ | $74.72_{\pm4.27}$ | $75.75_{\pm1.51}$ | $\mathbf{78.82}_{\pm1.74}$ | 3.07 | 0.76 | 4.44 |
| Citeseer | $\underline{73.89}_{\pm0.95}$ | $71.21_{\pm3.22}$ | $72.44_{\pm1.52}$ | $65.19_{\pm5.54}$ | $\mathbf{77.32}_{\pm2.42}$ | 4.88 | 6.11 | 3.43 |
| Pubmed | $\underline{51.98}_{\pm5.25}$ | $42.89_{\pm1.67}$ | $42.78_{\pm3.15}$ | $44.44_{\pm2.40}$ | $\mathbf{57.33}_{\pm2.42}$ | 12.89 | 14.44 | 5.35 |
| CS | $59.51_{\pm7.34}$ | $34.01_{\pm9.37}$ | $40.69_{\pm5.12}$ | $\underline{61.10}_{\pm2.83}$ | $\mathbf{68.62}_{\pm1.46}$ | 7.52 | 34.61 | 9.11 |
| Physics | $\underline{66.74}_{\pm1.53}$ | $31.26_{\pm9.12}$ | $52.11_{\pm2.44}$ | $52.34_{\pm3.78}$ | $\mathbf{72.01}_{\pm1.89}$ | 19.67 | 40.75 | 5.27 |
| Computers | $\underline{31.66}_{\pm3.08}$ | $20.19_{\pm1.58}$ | $12.81_{\pm1.80}$ | $21.75_{\pm1.96}$ | $\mathbf{35.32}_{\pm2.28}$ | 13.57 | 15.31 | 3.66 |
| Photos | $\mathbf{51.50}_{\pm4.48}$ | $27.83_{\pm4.90}$ | $24.24_{\pm2.79}$ | $38.47_{\pm2.76}$ | $\underline{49.32}_{\pm2.64}$ | 10.85 | 21.49 | -2.18 |
| OGB-Collab | $\mathbf{48.69}_{\pm0.87}$ | $36.95_{\pm1.37}$ | $35.97_{\pm0.96}$ | $36.86_{\pm0.45}$ | $\underline{45.27}_{\pm0.79}$ | 8.41 | 8.32 | -3.42 |
| OGB-Citation2 | $\mathbf{82.56}_{\pm0.04}$ | $40.63_{\pm0.00}$ | $38.42_{\pm0.01}$ | $42.50_{\pm0.01}$ | $\underline{53.20}_{\pm1.20}$ | 10.70 | 12.57 | -29.36 |

Chen et al., 2021b). Nonetheless, all three types of these edges appear with varying proportions in practical use-cases, e.g. growth of a social network or online platform, hence we evaluate on all three types. Note that we only conduct production setting experiments on non-OGB datasets, because the two OGB datasets are already temporally split in their public releases. We further elaborate the details of the production setting as well as the statistics in Appendix C.

For OGB datasets, we use their official metric (Hits@50 for `OGB-Collab` and Mean reciprocal Rank (MRR) for `OGB-Citation2`) following the public leaderboard[1]. For other datasets, following previous works (Yun et al., 2021; Zhang et al., 2021a; Zhao et al., 2022), we use Hits@20 as the main metric, which is also one of the main metrics on OGB datasets. We also report AUC performance in Appendix D. For all experiments, we report the averaged test performance (with early-stopping on validation) along with its standard deviation over 10 runs with different random initializations.

**Methods.** In the remainder of this section: "GNN" indicates the teacher GNN that was trained with $\mathcal{L}_{sup}$; "MLP" refers to the stand-alone MLP that was trained with $\mathcal{L}_{sup}$; "$\mathcal{L}_{LM}$" refers to MLP trained with logit matching (Equation (3)); "$\mathcal{L}_{RM}$" refers to MLP trained with node representation matching (Equation (4)); "LLP" refers to MLP trained with our proposed relational KD (Equation (7)). For the main experiments, we opt for simplicity and use SAGE (Hamilton et al., 2017) as the teacher GNN in all settings. We also include further experiments of different teacher GNN models in Figure 5 (in Appendix D.3).

## 4.2 LINK PREDICTION RESULTS

**Transductive Setting.** Table 1 shows the link prediction performance of the proposed LLP with GNN, MLP, and the direct KD methods (as introduced in Section 3.1) in the transductive setting. We observe that LLP consistently outperforms MLP and direct KD methods across all datasets with large margins. Specifically, LLP achieves **17.13** points and **10.17** points improvements over MLP and direct KD methods averaged on datasets, respectively. On the `Physics` dataset, LLP achieves **40.75** points and **19.67** points absolute improvements over MLP and direct KD, respectively. Moreover, LLP achieves better performance than the teacher GNN model on 6 out of 9 datasets, demonstrating that our proposed rank-based and distribution-based matching are able to effectively distill the relational knowledge for link prediction.

**Production Setting.** Table 2 shows the link prediction performance of the proposed LLP with GNN, MLP, and the direct KD methods in the production setting. For ease of comparison, we also stratify each method's performance on the three different categories of the test edges. We observe that LLP is still able to consistently outperform MLP and direct KD methods by large margins for all test categories. Specifically, LLP achieves **12.01** and **6.67** on Hits@20 improvements over MLP and direct KD methods averaged over datasets, respectively. Moreover, LLP is at or above par with the teacher GNN on 3 out of the 6 datasets. From the more stratified test performances, we observe that LLP can generally achieve similar performance with GNN on the existing–existing category, but much worse on the other two categories that involve newly appeared nodes. We hypothesize that this is because GNN neighbor aggregation improves generalization for low-degree nodes. We also

[1]https://ogb.stanford.edu/docs/leader_linkprop/

Table 2: The performance measured by Hits@20 of production setting. Best and second best performances are marked with bold and underline, respectively.

| | GNN | MLP | $\mathcal{L}_{LM}$ | $\mathcal{L}_{RM}$ | LLP | $\Delta_{DirectKD}$ | $\Delta_{MLP}$ | $\Delta_{GNN}$ |
|---|---|---|---|---|---|---|---|---|
| Overall | | | | | | | | |
| Cora | $\underline{27.80}_{\pm2.11}$ | $22.90_{\pm2.22}$ | $22.65_{\pm2.51}$ | $22.24_{\pm0.55}$ | $\mathbf{27.87}_{\pm1.24}$ | 5.22 | 4.97 | 0.07 |
| Citeseer | $\mathbf{38.78}_{\pm2.59}$ | $31.21_{\pm3.75}$ | $29.35_{\pm2.55}$ | $26.23_{\pm1.08}$ | $\underline{34.75}_{\pm2.45}$ | 5.40 | 3.54 | -4.03 |
| Pubmed | $\underline{52.71}_{\pm1.81}$ | $38.01_{\pm1.67}$ | $39.03_{\pm4.21}$ | $43.27_{\pm3.12}$ | $\mathbf{53.48}_{\pm1.52}$ | 10.21 | 15.47 | 0.77 |
| CS | $\underline{60.69}_{\pm3.17}$ | $38.15_{\pm10.78}$ | $48.07_{\pm2.39}$ | $58.90_{\pm1.32}$ | $\mathbf{60.74}_{\pm1.41}$ | 1.84 | 22.59 | 0.05 |
| Physics | $\mathbf{55.82}_{\pm2.43}$ | $29.99_{\pm1.96}$ | $22.74_{\pm1.03}$ | $36.32_{\pm2.29}$ | $\underline{52.83}_{\pm1.50}$ | 16.51 | 22.84 | -2.99 |
| Computers | $\mathbf{34.38}_{\pm1.41}$ | $19.43_{\pm0.82}$ | $12.79_{\pm1.43}$ | $20.28_{\pm1.01}$ | $\underline{24.58}_{\pm3.33}$ | 4.30 | 5.15 | -9.80 |
| Photos | $\mathbf{51.03}_{\pm6.05}$ | $34.29_{\pm2.49}$ | $24.63_{\pm2.20}$ | $40.58_{\pm1.63}$ | $\underline{43.79}_{\pm1.27}$ | 3.21 | 9.50 | -7.24 |
| Existing – Existing | | | | | | | | |
| Cora | $\underline{28.81}_{\pm2.01}$ | $28.00_{\pm2.70}$ | $27.66_{\pm3.01}$ | $27.03_{\pm0.65}$ | $\mathbf{33.31}_{\pm1.29}$ | 5.65 | 5.31 | 4.5 |
| Citeseer | $\mathbf{38.10}_{\pm2.70}$ | $33.88_{\pm3.50}$ | $32.24_{\pm2.89}$ | $27.52_{\pm0.94}$ | $\underline{37.50}_{\pm2.43}$ | 5.26 | 3.62 | -0.60 |
| Pubmed | $\underline{52.67}_{\pm1.78}$ | $41.58_{\pm1.61}$ | $42.57_{\pm4.32}$ | $46.32_{\pm3.08}$ | $\mathbf{57.16}_{\pm1.34}$ | 10.84 | 15.58 | 4.49 |
| CS | $\underline{61.52}_{\pm3.10}$ | $40.27_{\pm11.69}$ | $50.78_{\pm2.50}$ | $62.17_{\pm1.45}$ | $\mathbf{63.99}_{\pm1.36}$ | 1.82 | 23.72 | 2.47 |
| Physics | $\mathbf{56.56}_{\pm2.42}$ | $32.32_{\pm2.32}$ | $23.88_{\pm1.14}$ | $38.74_{\pm2.50}$ | $\underline{56.04}_{\pm1.47}$ | 17.30 | 23.72 | -0.52 |
| Computers | $\mathbf{35.13}_{\pm1.48}$ | $21.46_{\pm1.08}$ | $13.81_{\pm1.56}$ | $22.78_{\pm1.17}$ | $\underline{26.89}_{\pm3.60}$ | 4.11 | 5.43 | -8.24 |
| Photos | $\mathbf{51.90}_{\pm6.24}$ | $37.47_{\pm2.73}$ | $26.54_{\pm2.55}$ | $44.51_{\pm2.10}$ | $\underline{48.38}_{\pm1.30}$ | 3.87 | 10.91 | -3.52 |
| Existing – New | | | | | | | | |
| Cora | $\mathbf{25.78}_{\pm2.33}$ | $19.47_{\pm2.09}$ | $19.11_{\pm2.03}$ | $18.58_{\pm1.28}$ | $\underline{23.08}_{\pm1.51}$ | 3.97 | 3.61 | -2.7 |
| Citeseer | $\mathbf{38.73}_{\pm2.37}$ | $30.77_{\pm4.07}$ | $28.77_{\pm2.70}$ | $26.65_{\pm1.52}$ | $\underline{34.30}_{\pm2.40}$ | 5.53 | 3.53 | -4.43 |
| Pubmed | $\mathbf{53.98}_{\pm2.29}$ | $23.70_{\pm2.09}$ | $24.91_{\pm4.00}$ | $32.21_{\pm3.38}$ | $\underline{38.94}_{\pm2.44}$ | 6.73 | 15.24 | -15.04 |
| CS | $\mathbf{56.78}_{\pm3.57}$ | $29.25_{\pm7.05}$ | $36.60_{\pm2.17}$ | $45.28_{\pm0.93}$ | $\underline{47.05}_{\pm1.72}$ | 1.77 | 17.80 | -9.73 |
| Physics | $\mathbf{52.90}_{\pm2.44}$ | $20.61_{\pm1.01}$ | $18.23_{\pm0.74}$ | $26.57_{\pm1.81}$ | $\underline{39.73}_{\pm1.75}$ | 13.16 | 19.12 | -13.17 |
| Computers | $\mathbf{31.07}_{\pm1.17}$ | $11.00_{\pm1.37}$ | $8.53_{\pm1.16}$ | $9.85_{\pm0.54}$ | $\underline{14.88}_{\pm2.58}$ | 5.03 | 3.88 | -16.19 |
| Photos | $\mathbf{47.42}_{\pm5.18}$ | $21.00_{\pm1.65}$ | $16.75_{\pm0.92}$ | $24.10_{\pm1.38}$ | $\underline{24.27}_{\pm2.07}$ | 0.17 | 3.27 | -23.15 |
| New – New | | | | | | | | |
| Cora | $\mathbf{31.97}_{\pm6.65}$ | $11.69_{\pm2.19}$ | $12.54_{\pm2.83}$ | $13.80_{\pm1.37}$ | $\underline{16.90}_{\pm5.50}$ | 3.10 | 5.21 | -15.07 |
| Citeseer | $\mathbf{42.74}_{\pm4.49}$ | $18.71_{\pm4.54}$ | $16.29_{\pm3.80}$ | $17.26_{\pm3.54}$ | $\underline{21.94}_{\pm4.39}$ | 4.68 | 3.23 | -20.8 |
| Pubmed | $\mathbf{33.18}_{\pm1.24}$ | $5.45_{\pm1.24}$ | $4.55_{\pm4.55}$ | $11.36_{\pm4.82}$ | $\underline{15.00}_{\pm6.35}$ | 3.64 | 9.55 | -18.18 |
| CS | $\mathbf{64.10}_{\pm3.55}$ | $26.27_{\pm8.79}$ | $33.73_{\pm3.81}$ | $38.07_{\pm2.90}$ | $\underline{42.89}_{\pm1.83}$ | 4.82 | 16.62 | -21.21 |
| Physics | $\mathbf{48.96}_{\pm3.53}$ | $13.20_{\pm1.62}$ | $12.56_{\pm1.85}$ | $18.88_{\pm2.22}$ | $\underline{32.80}_{\pm1.55}$ | 13.92 | 19.6 | -16.16 |
| Computers | $\mathbf{32.61}_{\pm1.89}$ | $5.55_{\pm1.56}$ | $6.72_{\pm0.66}$ | $3.87_{\pm1.24}$ | $\underline{10.25}_{\pm1.41}$ | 3.53 | 4.7 | -22.36 |
| Photos | $\mathbf{43.54}_{\pm6.6}$ | $10.09_{\pm3.99}$ | $8.14_{\pm1.15}$ | $12.21_{\pm1.31}$ | $\underline{14.87}_{\pm3.09}$ | 2.66 | 4.78 | -28.67 |

Figure 2: Inference time comparison of LLP with GNN inference acceleration methods in log scale.

Figure 3: Link prediction performance measured by Hits@20 of LLP on Pubmed with different number of samples for the context nodes.

observe that the performance gaps between the teacher GNN and stand-alone MLP on new–new and existing–new are much larger than that of the existing–existing category, which also evidences that GNNs have better inductive bias than MLPs on graph data. Nonetheless, we note that such a significant and consistent performance improvements of LLP over MLP is valuable for large-scale industrial applications, given their popularity in practice.

## 4.3 INFERENCE ACCELERATION COMPARISON

We evaluate LLP in comparison to other common GNN inference acceleration methods, which mainly focus on the hardware and algorithm to reduce the computation consuming, such as pruning (Zhou et al., 2021) and quantization (Zhao et al., 2020; Tailor et al., 2020). Following the experimental

settings in Zhang et al. (2021b), we measure the inductive inference time on 10 randomly chosen nodes in the graph. We evaluate against 4 common GNN inference acceleration methods: (i) SAGE (Hamilton et al., 2017), (ii) Quantized SAGE (QSAGE) (Zhao et al., 2020; Tailor et al., 2020) from `float32` to `int8`, (iii) SAGE with 50% weights pruned (PSAGE) (Zhou et al., 2021; Chen et al., 2021c), and (iv) SAGE with Neighbor Sampling with fan-out 15. Figure 2 shows the results on the large-scale OGB datasets. We can observe that LLP can obtain **71.04×** and **776.37×** speedup comparing with on SAGE on `OGB-Collab` and `OGB-Citation2` datasets, respectively. Comparing with the best acceleration method Neighbor Sampling (which reduces graph dependency, but does not eliminate it like LLP), LLP still achieves **15.12×** and **50.51×** speedup on the two datasets, respectively. This is because all these inference acceleration methods still rely on the graph structure.

### 4.4 LINK PREDICTION RESULTS ON COLD START NODES

Dealing with cold start nodes (newly appeared nodes without edges) is a common challenge in recommendation and information retrieval applications (Li et al., 2019; Zheng et al., 2021; Ding et al., 2021). Without these edges, GNNs cannot perform well as they rely heavily on neighbor information. On the other hand, MLPs, which do not make use of any graph topology information, are arguably more suitable. Here, we simulate

Table 3: Link pred. Hits@20 on cold-start nodes.

| | GNN | MLP | Ours | $\Delta_{MLP}$ | $\Delta_{GNN}$ |
|---|---|---|---|---|---|
| Cora | 6.39 | 17.92 | **22.01** | 4.09 | 15.62 |
| Citeseer | 11.04 | 29.33 | **32.09** | 2.76 | 21.05 |
| Pubmed | 4.63 | 22.74 | **37.68** | 14.94 | 33.05 |
| CS | 9.46 | 29.09 | **46.83** | 17.74 | 37.37 |
| Physics | 5.46 | 20.22 | **39.37** | 19.15 | 33.91 |
| Computers | 1.53 | 10.72 | **14.64** | 3.92 | 13.11 |
| Photos | 0.87 | 20.44 | **23.79** | 3.35 | 22.92 |

the cold-start setting by removing all the new edges during testing stage of the production setting, i.e. all the new nodes are isolated (see Appendix C for more details). Table 3 shows the performances of LLP, the stand-alone MLP, and the teacher GNN on the cold-start nodes. We observe that LLP consistently outperforms GNN and MLP by average of **25.29** and **9.42** on Hits@20, respectively.

### 4.5 ABLATION STUDY

**Effectiveness of $\mathcal{L}_{\text{LLP}\_R}$ and $\mathcal{L}_{\text{LLP}\_D}$.** As our proposed LLP contains two matching strategies, rank-based and distribution-based matching, we evaluate their effectiveness by removing them from LLP. Moreover, we further evaluate by also removing $\mathcal{L}_{sup}$, i.e., using only one of the matching losses as the overall loss for LLP. Table 4 shows the results of these settings compared with the performances of full LLP, stand-alone MLP, and the teacher GNN on Pubmed and CS datasets under both settings. We observe that both rank-

Table 4: Results measured by Hits@20 on different components of LLP.

| Setting | Transductive | | Production | |
|---|---|---|---|---|
| Dataset | Pubmed | CS | Pubmed | CS |
| GNN | 51.98 | 59.51 | 52.71 | 60.69 |
| MLP | 42.89 | 40.69 | 38.01 | 38.15 |
| LLP | **57.33** | **68.62** | **53.48** | **60.74** |
| w/o $\mathcal{L}_{\text{LLP}\_R}$ | 55.35 | 66.61 | 53.40 | 60.53 |
| w/o $\mathcal{L}_{\text{LLP}\_D}$ | 54.97 | 65.17 | 48.58 | 60.13 |
| w/o $\mathcal{L}_{\text{LLP}\_R}, \mathcal{L}_{sup}$ | 54.86 | 68.39 | 39.35 | 57.35 |
| w/o $\mathcal{L}_{\text{LLP}\_D}, \mathcal{L}_{sup}$ | 53.30 | 68.30 | 41.43 | 55.63 |

based and distribution-based matching contribute significantly for the overall performance. In the transductive setting, both loss terms by themselves (the bottom two rows) can already outperform the teacher GNN. In the production setting, the matching losses alone outperform MLP and can achieve comparable performances with GNN after $\mathcal{L}_{sup}$ is added. In conclusion, both rank-based and distribution-based matching can effectively distill the relational knowledge, and they achieve the best performance by complementing each other.

**Context Sampling Sensitivity to $p$ and $q$.** Figure 3 shows the link prediction performance of LLP on Pubmed under the transductive setting with different numbers of context node samples ($p$ local samples, and $q$ random samples). For the ease of hyper-parameter tuning, we make $q$ a multiple of $p$, as shown in the $x$-axis of Figure 3. We observe that low number of random samples show poor link prediction performances, suggesting that preserving global relations are necessary for the proposed relation KD. Generally, the heatmap shows a clear trend, making the optimal values easy to locate.

## 5 CONCLUSION

Our work tackled problems related to applying GNNs for link prediction at scale. We note these models have high latency at inference time owing to non-trivial data dependency. In response, we explored applying cross-model distillation methods from teacher GNN to student MLP models, which are advantaged in inference time. We first adopt two direct logit matching and representation matching KD methods to the link prediction context and observe their unsuitability. In response, we introduced

a relational KD framework, LLP, which proposed *rank-based matching* and *distribution-based matching* objectives which complement each other to force the student to preserve key information about contextual relationships across anchor nodes. Our experiments demonstrated that LLP achieved MLP-level speedups (up to **776.37×** over GNNs), while also improving link prediction performance over MLPs by **17.13** and **12.01** points in transductive and production settings, matching or outperforming the teacher GNN in 6/9 datasets in transductive setting and 3/9 datasets in production setting, and with notable **25.29** on Hits@20 improvements on cold-start nodes.

## REPRODUCIBILITY STATEMENT

To ensure the reproducibility of our experiments, we provide the source code for LLP in the supplementary materials. Moreover, we will publically open-source our code later after we clean up our code package and add proper documentation for it. The hyper-parameters that are required to reproduce our experiments are provided in Appendix E.

## ACKNOWLEDGEMENT

We appreciate Xiaotian Han from Texas A&M University, Wei Jin from Michigan State University, and Yiwei Wang from National University of Singapore for valuable discussions and suggestions.

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

# A  FURTHER RELATED WORK

In this section, we discuss other work related to LLP.

**Graph Neural Networks (GNNs).** Many GNN architectures have been proposed in recent years to model attributed graph data; most architectures follow the message passing (Gilmer et al., 2017; Guo et al., 2021; Ma et al., 2021; Liu et al., 2021; 2022) paradigm. Different GNN customizations include degree normalization (Kipf & Welling, 2016a), neighbor sampling and neighbor separation (Hamilton et al., 2017; Zhao et al., 2021b), self-attention (Veličković et al., 2017), residual connections (Xu et al., 2018), and more. Alon & Yahav (2020) proposed to use a fully-adjacent layer at the end of GNN to deal with the bottleneck problem of GNNs. Moreover, researchers also proposed subgraph-based methods (Bevilacqua et al., 2021; Zhao et al., 2021a), and tensor-based methods (Maron et al., 2019; Geerts & Reutter, 2022) for more expressive GNNs.

**Link Prediction.** Link prediction has achieved great attention from the research community, considering its wide applications. Heuristic methods (Philip et al., 2010) were proposed to make the link prediction by measuring the link scores based on the structure information, such as the common neighbors and the shortest path. 2-order (Adamic & Adar, 2003) and high-order (Brin & Page, 2012; Jeh & Widom, 2002) heuristic methods were proposed to further improve the effectiveness. Link prediction with GNN is another important direction for link prediction (Yun et al., 2021), which is based on the learned embeddings. One line of work is the strategy we discussed in Section 2, where the GNN-based encoder learns node representations and the decoder predicts whether the link exists. It is worth mentioning that knowledge graph completion follows this strategy to predict not only the link existence but also the type of the link (Schlichtkrull et al., 2018; Nathani et al., 2019; Vashishth et al., 2020; Zhang et al., 2020a). These methods mainly use heterogeneous graph neural networks sensitive to different edge types. Another line of work casts link prediction tasks as classification problems on the enclosing subgraphs around each link (Zhang & Chen, 2018; Cai & Ji, 2020; Cai et al., 2021). Although these methods can improve task performance, they are usually extremely computationally expensive and cannot scale well in practical use-cases (Yin et al., 2022). Similarly, Zhu et al. (2021) proposed a GNN link prediction paradigm by encoding information of all paths between two nodes, which is also very expensive. Our work focuses on accelerating the more general and faster methods discussed in Section 2.

**GNN Inference Acceleration.** Pruning (Zhou et al., 2021; Chen et al., 2021c) and quantization (Zhao et al., 2020; Tailor et al., 2020) strategies were proposed for accelerating GNN inference. These methods do accelerate GNNs, but they rely on graph data for message passing and thus leave much room for speed improvement. We note that these approaches are complementary to cross-model distillation, and can be employed together with KD for additional inference time improvements. Other than the above acceleration methods, Hu et al. (2021) and Zhang et al. (2021b) accelerated GNNs by distilling them to MLP. These works focus on KD for node classification tasks, whereas we focus on link prediction tasks. GNNAutoScale (Fey et al., 2021) proposed an effective method to accelerate the training process of GNNs. It also reduces the inference to a constant factor by directly using historical embeddings stored offline. However, in this case, all the methods in Figure 2 can share the same inference time benefits. Moreover, GNNAutoScale is not suitable for the production setting, where new nodes (without historical embeddings) appear frequently after the training process. So we did not include it as a baseline in this work.

**Knowledge Distillation (KD).** Logit-based (Hinton et al., 2015; Furlanello et al., 2018; Zhang et al., 2021b) and representation-based (Romero et al., 2014; Gou et al., 2021) matching are two common KD methods, which match final-layer and intermediate-layer predicted logits between the teacher and the student, respectively. Our work is the first to adapt and evaluate these approaches in the link prediction setting, to the best of our knowledge.

For representation based KD, several work (Park et al., 2019; Tung & Mori, 2019; Joshi et al., 2021) proposed relational KD, which corresponds to instance-to-instance KD while preserving metrics among representations of similar instances. For GNNs, Yang et al. (2020a) used knowledge of the neighboring nodes to teach the student to better classify the center node. In contrast, our KD strategies focus on transferring relational knowledge between each pair of nodes from teacher to student. Both the rank-matching and distribution-matching strategies help the student to better capture the relational graph topology information and make better link prediction.

Table 5: Statistics of all datasets used in the experiments.

| Dataset | # Nodes | # Edges | # Features |
|---|---|---|---|
| Cora | 2,708 | 5.278 | 1,433 |
| Citeseer | 3,327 | 4,552 | 3,703 |
| Pubmed | 19,717 | 44,324 | 500 |
| CS | 18,333 | 163,788 | 6,805 |
| Physics | 34,493 | 495,924 | 8,415 |
| Computers | 13,752 | 491,722 | 767 |
| Photos | 7,650 | 238,162 | 745 |
| OGB-Collab | 235,868 | 1,285,465 | 128 |
| OGB-Citation2 | 2,927,963 | 30,561,187 | 128 |

Table 6: Detailed statistics of data splits under production setting.

| | Nodes | | Testing Edges | | |
| | # Existing | # New | # Existing – Existing | # Existing – New | # New – New |
|---|---|---|---|---|---|
| Cora | 1,896 | 812 | 765 | 675 | 142 |
| Citeseer | 2,329 | 998 | 673 | 568 | 124 |
| Pubmed | 15,774 | 3,943 | 5,648 | 2,858 | 358 |
| CS | 14,666 | 3,667 | 10,482 | 5,221 | 675 |
| Physics | 27,594 | 6,899 | 31,399 | 16,126 | 2,067 |
| Computers | 11,002 | 2,750 | 31,095 | 16,033 | 2,043 |
| Photos | 6,120 | 1,530 | 15,248 | 7,618 | 950 |

RankDistill (Reddi et al., 2021) is designed to transfer ranking knowledge from the teacher to the student. Different from our work which distill the relational information in a graph context, it distills ranking in a non-graph context between teacher and student. We adopt different sampling and matching methods based on our different motivations. Further analysis is shown in Appendix D.9.

**KD on GNNs.** Existing GNN-based KD work are mostly based on the logit-based KD (Hinton et al., 2015) to obtain light-weight models (Zhang et al., 2020d; Zheng et al., 2021; Yang et al., 2021). Yan et al. (2020) proposed to train a student GNN with fewer parameters using KD. Yang et al. (2021) improved the designed student model, which consists of label propagation and feature-based prior knowledge, using the pre-trained teacher GNN. Different from the above work, LSP (Yang et al., 2020a) and G-CRD (Joshi et al., 2021) proposed structure-preserving KD methods, which are specifically designed for GNN. Both of these work follow the original relational KD to preserve the metrics among node representations and are applied on node classification tasks.

## B    ADDITIONAL DATASETS DETAILS

Here we present the details of the datasets used in the experiments. Cora, Citeseer, Pubmed (Yang et al., 2016) and OGB-Citation2 (Wang et al., 2020a; Mikolov et al., 2013) are all representative citation network datasets, where the nodes and edges represent papers and citations, respectively. CS, Physics (Shchur et al., 2018) and OGB-Collab (Wang et al., 2020a) are all collaboration networks based on MAG, where the nodes represent authors and the edges indicate the collaboration for the paper. Computers and Photos (Shchur et al., 2018) are two well-known co-purchased graphs (McAuley et al., 2015), where the nodes represent goods and the edges indicate two items were bought together. The detailed statistics of these datasets are shown in Table 5.

## C  ADDITIONAL EVALUATION SETTING DETAILS

### C.1  TRANSDUCTIVE SETTING

The transductive setting is a standard setting for link prediction (Kipf & Welling, 2016b; Zhang & Chen, 2017; 2018; Yun et al., 2021; Zhao et al., 2022), where the nodes in training/validation/testing are all visible in the training graph, but subsets of positive links are masked out for validation and test sets.

### C.2  PRODUCTION SETTING

In this work, we design a new production setting to resemble the real-world link prediction scenario. This setting mimics pratical link prediction use-cases. For example, user friend recommendation on social platforms where new users (nodes) and friendships (links) appear frequently. Under the production setting, the newly occurred nodes and edges that can not be seen during the training stage would appear in the graph at inference time.

Specifically, following are the detailed procedures of splitting the datasets into the production setting:

- **Split all nodes:** Given the graph $G = (\mathcal{V}, \mathcal{E})$, we randomly sample 10% of nodes from $\mathcal{V}$ as the new nodes $\mathcal{V}^N$ and remove them from the training graph. We denote the remaining nodes by $\mathcal{V}^E$, where superscripts $E$ stands for *Existing* and $N$ stands for *New*. Note that for Cora and Citeseer, we sample 30% nodes as new nodes because these two datasets are too small.

- **Split all edges:** We then split the edges $\mathcal{E}$ according to the node splits into three sets: $\mathcal{E}^{E-E}$, $\mathcal{E}^{E-N}$, and $\mathcal{E}^{N-N}$, denoting the links between existing–existing, existing–new, and new–new node pairs, respectively.

- **Split edges in $\mathcal{E}^{E-E}$:** For the existing–existing node pairs, we split it into three sets following an 80/10/10 splitting ratio: 80% as training edges, 10% as new visible edges for message passing, and 10% as testing edges. Note that validation set contains only existing nodes $\mathcal{V}^E$ as the new nodes are not visible during training.

- **Split edges in $\mathcal{E}^{E-N}$ and $\mathcal{E}^{N-N}$:** We follow the same ratio and split these two sets following with 90/10 splitting ratio: 90% as newly visible edges (used only for message passing during testing inference), and 10% as testing edges.

- **Message passing edges during training:** During training, the GNN model can only utilize the 80% exising-existing training edges for message passing.

- **Message passing edges for inferencing:** During inference, the GNN model can conduct message passing on all exist except the testing ones. Specifically, the training and testing (total of 90%) sets of $\mathcal{E}^{E-E}$, and the 90% of newly visible message passing edges in $\mathcal{E}^{E-N}$ and $\mathcal{E}^{N-N}$.

- **Testing edges:** We test all methods on the above-mentioned three separate testing edge sets (10% of each) sampled from $\mathcal{E}^{E-E}$, $\mathcal{E}^{E-N}$, and $\mathcal{E}^{N-N}$, respectively.

Table 6 shows the detailed statistic of different datasets under this setting.

### C.3  COLD-START SETTING

Followed by the production setting, we remove all the new edges appearing newly in the inference time. Then the new nodes will be the strict cold start nodes with no neighbor information for the model to predict. The experimental results showed in Section 4.4 are conducted with this setting.

## D  ADDITIONAL EXPERIMENTAL RESULTS

### D.1  AUC RESULTS ON TRANSDUCTIVE AND PRODUCTION SETTINGS

Here, we present AUC results on all the non-OGB datasets under the transductive setting in Table 7 and production setting in Table 8. In Table 7, we can observe that our method outperforms both MLP

Table 7: Link prediction performance measured by AUC under transductive setting.

| | GNN | MLP | $\mathcal{L}_{LM}$ | $\mathcal{L}_{RM}$ | LLP | $\Delta_{DirectKD}$ | $\Delta_{MLP}$ | $\Delta_{GNN}$ |
|---|---|---|---|---|---|---|---|---|
| Cora | $95.03_{\pm0.37}$ | $94.80_{\pm0.44}$ | $94.67_{\pm0.58}$ | $94.05_{\pm0.17}$ | $\mathbf{95.23}_{\pm0.49}$ | 0.56 | 0.43 | 0.20 |
| Citeseer | $95.15_{\pm0.58}$ | $93.11_{\pm1.21}$ | $94.11_{\pm0.21}$ | $92.88_{\pm0.37}$ | $\mathbf{95.32}_{\pm0.21}$ | 1.21 | 2.21 | 0.17 |
| Pubmed | $93.84_{\pm0.31}$ | $97.89_{\pm0.07}$ | $97.82_{\pm0.06}$ | $\mathbf{97.96}_{\pm0.02}$ | $97.90_{\pm0.09}$ | -0.06 | 0.01 | 4.06 |
| CS | $97.43_{\pm0.23}$ | $97.61_{\pm0.52}$ | $98.05_{\pm0.14}$ | $\mathbf{98.33}_{\pm0.05}$ | $98.06_{\pm0.04}$ | -0.27 | 0.45 | 0.63 |
| Physics | $98.80_{\pm0.02}$ | $98.71_{\pm0.05}$ | $98.36_{\pm0.07}$ | $98.96_{\pm0.02}$ | $\mathbf{99.10}_{\pm0.02}$ | 0.14 | 0.39 | 0.30 |
| Computers | $98.76_{\pm0.03}$ | $98.46_{\pm0.08}$ | $98.11_{\pm0.14}$ | $98.66_{\pm0.06}$ | $\mathbf{98.84}_{\pm0.09}$ | 0.18 | 0.38 | 0.08 |
| Photos | $98.98_{\pm0.02}$ | $98.71_{\pm0.08}$ | $98.51_{\pm0.06}$ | $98.95_{\pm0.04}$ | $\mathbf{99.03}_{\pm0.06}$ | 0.08 | 0.32 | 0.05 |

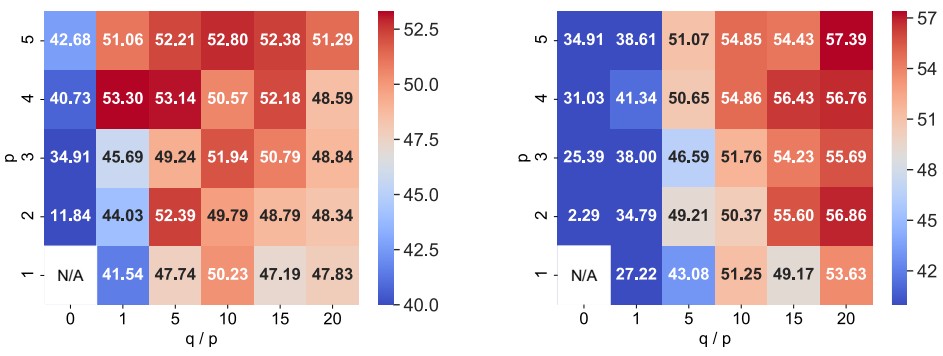

Figure 4: Link prediction performance measured by Hits@20 of $\mathcal{L}_{\mathsf{LLP\_R}}$ (left) and $\mathcal{L}_{\mathsf{LLP\_D}}$ (right) on Pubmed with different $p$ and $q$.

and the teacher GNN on all the datasets under the transductive setting. For the production setting, our method performs better than the teacher GNN on 4/7 datasets, as shown in Table 8.

## D.2 SENSITIVITY ANALYSIS OF $p$ AND $q$ FOR $\mathcal{L}_{\mathsf{LLP\_R}}$ AND $\mathcal{L}_{\mathsf{LLP\_D}}$

To analyze the influence of the context nodes on $\mathcal{L}_{\mathsf{LLP\_R}}$ and $\mathcal{L}_{\mathsf{LLP\_D}}$, we plot two heat maps to show their individual performance on Pubmed under the transductive setting, as shown in Figure 4. These two figures show different patterns with the context nodes. $\mathcal{L}_{\mathsf{LLP\_D}}$ (left figure) shows that the performance becomes better with more nearby nodes ($p$) and a higher random sampled rate($q/p$). And random sampling rate can lead to a much better performance than nearby nodes. However, in $\mathcal{L}_{\mathsf{LLP\_R}}$, we find that the results on the diagonal perform consistently better than those around, which means the random sampling rate should match the nearby nodes to work well. Besides, we can also observe that $\mathcal{L}_{\mathsf{LLP\_R}}$ is more sensitive with a smaller number of context nodes than $\mathcal{L}_{\mathsf{LLP\_D}}$. $\mathcal{L}_{\mathsf{LLP\_R}}$ matches the performance by the relative ranking of the context nodes w.r.t. the anchor nodes. However, it becomes difficult for $\mathcal{L}_{\mathsf{LLP\_R}}$ to learn well when there are many context nodes. In contrast, $\mathcal{L}_{\mathsf{LLP\_D}}$ matches the distribution, and more context nodes provides a clearer picture about the link-related structure around the anchor node.

## D.3 RESULTS WITH DIFFERENT TEACHERS

In our experiments, we use SAGE as the teacher GNN in both transductive and production settings. We further test LLP's performance with other GNNs, such as GCN, GAT, and APPNP. In Figure 5, we find that our model always outperforms MLP with different GNN teachers. However, absolute performance is closely related to the teacher.

## D.4 IMPORTANCE ANALYSIS OF $\mathcal{L}_{\mathsf{LLP\_R}}$, $\mathcal{L}_{\mathsf{LLP\_D}}$ AND $\mathcal{L}_{sup}$ FOR LLP

We conduct the ablation study on all non-OGB datasets to analyze the contributions of each component in Equation (7). In each ablation setting, we remove one component independently, as shown in Figure 6. We can observe that the performance drops when any of the three components (i.e.,

Table 8: Link prediction performance measured by AUC under production setting.

| | GNN | MLP | $\mathcal{L}_{LM}$ | $\mathcal{L}_{RM}$ | LLP | $\Delta_{DirectKD}$ | $\Delta_{MLP}$ | $\Delta_{GNN}$ |
|---|---|---|---|---|---|---|---|---|
| Overall | | | | | | | | |
| Cora | $72.59_{\pm1.63}$ | $\underline{73.41}_{\pm2.04}$ | $70.67_{\pm1.62}$ | $64.62_{\pm0.51}$ | $\mathbf{78.22}_{\pm1.14}$ | 7.55 | 4.81 | 5.63 |
| Citeseer | $69.15_{\pm1.82}$ | $\underline{77.36}_{\pm3.38}$ | $75.04_{\pm3.20}$ | $67.67_{\pm0.59}$ | $\mathbf{80.13}_{\pm0.98}$ | 5.09 | 2.77 | 10.98 |
| Pubmed | $90.45_{\pm0.45}$ | $96.07_{\pm0.13}$ | $\underline{96.13}_{\pm0.26}$ | $\mathbf{96.74}_{\pm0.05}$ | $94.30_{\pm0.34}$ | -2.44 | -1.77 | 3.85 |
| CS | $\mathbf{97.08}_{\pm0.16}$ | $95.96_{\pm1.19}$ | $96.59_{\pm0.08}$ | $96.76_{\pm0.03}$ | $96.87_{\pm0.03}$ | 0.11 | 0.91 | -0.21 |
| Physics | $\underline{98.60}_{\pm0.02}$ | $97.70_{\pm0.04}$ | $97.46_{\pm0.08}$ | $98.00_{\pm0.01}$ | $\mathbf{98.75}_{\pm0.11}$ | 0.75 | 1.05 | 0.15 |
| Computers | $\mathbf{98.67}_{\pm0.05}$ | $97.85_{\pm0.04}$ | $97.59_{\pm0.07}$ | $\underline{97.95}_{\pm0.03}$ | $97.89_{\pm0.04}$ | -0.06 | 0.04 | -0.78 |
| Photos | $\mathbf{98.78}_{\pm0.14}$ | $97.97_{\pm0.08}$ | $97.85_{\pm0.06}$ | $\underline{98.18}_{\pm0.04}$ | $98.05_{\pm0.03}$ | -0.13 | 0.08 | -0.73 |
| Existing – Existing | | | | | | | | |
| Cora | $70.80_{\pm2.14}$ | $\underline{74.42}_{\pm2.70}$ | $70.69_{\pm2.00}$ | $64.82_{\pm0.75}$ | $\mathbf{78.43}_{\pm1.44}$ | 7.74 | 4.01 | 7.63 |
| Citeseer | $67.34_{\pm1.81}$ | $\underline{76.83}_{\pm3.41}$ | $73.79_{\pm3.12}$ | $68.00_{\pm2.03}$ | $\mathbf{78.36}_{\pm1.41}$ | 4.57 | 1.53 | 11.02 |
| Pubmed | $90.44_{\pm0.46}$ | $96.69_{\pm0.13}$ | $\underline{96.72}_{\pm0.21}$ | $\mathbf{97.24}_{\pm0.05}$ | $95.17_{\pm0.33}$ | -2.07 | -1.52 | 4.73 |
| CS | $\mathbf{97.01}_{\pm0.16}$ | $96.08_{\pm1.11}$ | $96.70_{\pm0.08}$ | $96.91_{\pm0.03}$ | $\underline{97.00}_{\pm0.03}$ | 0.09 | 0.92 | -0.01 |
| Physics | $\underline{98.60}_{\pm0.02}$ | $97.96_{\pm0.05}$ | $97.65_{\pm0.09}$ | $98.20_{\pm0.02}$ | $\mathbf{98.76}_{\pm0.16}$ | 0.56 | 0.80 | 0.16 |
| Computers | $\mathbf{98.70}_{\pm0.05}$ | $98.27_{\pm0.05}$ | $97.95_{\pm0.09}$ | $98.41_{\pm0.03}$ | $\underline{98.51}_{\pm0.04}$ | 0.10 | 0.24 | -0.19 |
| Photos | $\mathbf{98.80}_{\pm0.14}$ | $98.33_{\pm0.09}$ | $98.20_{\pm0.07}$ | $98.57_{\pm0.07}$ | $\underline{98.61}_{\pm0.04}$ | 0.04 | 0.28 | -0.19 |
| Existing – New | | | | | | | | |
| Cora | $\underline{72.61}_{\pm1.50}$ | $72.06_{\pm1.55}$ | $70.18_{\pm1.41}$ | $64.07_{\pm0.58}$ | $\mathbf{77.65}_{\pm1.12}$ | 7.47 | 5.59 | 5.04 |
| Citeseer | $69.90_{\pm1.88}$ | $\underline{77.58}_{\pm3.48}$ | $76.05_{\pm3.51}$ | $67.13_{\pm1.74}$ | $\mathbf{81.23}_{\pm0.71}$ | 5.18 | 3.65 | 11.33 |
| Pubmed | $90.82_{\pm0.38}$ | $93.67_{\pm0.23}$ | $\underline{93.82}_{\pm0.54}$ | $\mathbf{94.83}_{\pm0.14}$ | $90.97_{\pm0.74}$ | -3.86 | -2.70 | 0.15 |
| CS | $\mathbf{97.31}_{\pm0.20}$ | $95.46_{\pm1.53}$ | $96.18_{\pm0.12}$ | $96.18_{\pm0.08}$ | $\underline{96.31}_{\pm0.10}$ | 0.13 | 0.85 | -1.00 |
| Physics | $\mathbf{98.57}_{\pm0.04}$ | $96.64_{\pm0.08}$ | $96.66_{\pm0.09}$ | $\underline{97.17}_{\pm0.04}$ | $95.72_{\pm0.27}$ | -1.45 | -0.92 | -2.85 |
| Computers | $\mathbf{98.60}_{\pm0.05}$ | $\underline{96.23}_{\pm0.07}$ | $96.22_{\pm0.07}$ | $96.19_{\pm0.08}$ | $95.42_{\pm0.08}$ | -0.80 | -0.81 | -3.18 |
| Photos | $\mathbf{98.69}_{\pm0.14}$ | $96.53_{\pm0.03}$ | $96.45_{\pm0.09}$ | $\underline{96.60}_{\pm0.08}$ | $95.76_{\pm0.16}$ | -0.84 | -0.77 | -2.93 |
| New – New | | | | | | | | |
| Cora | $\mathbf{82.10}_{\pm1.57}$ | $74.46_{\pm1.40}$ | $72.85_{\pm1.92}$ | $66.12_{\pm1.39}$ | $\underline{79.85}_{\pm1.30}$ | 7.00 | 5.39 | -2.25 |
| Citeseer | $75.48_{\pm1.67}$ | $\underline{79.23}_{\pm3.08}$ | $77.13_{\pm3.24}$ | $68.36_{\pm2.67}$ | $\mathbf{84.68}_{\pm0.89}$ | 7.55 | 5.45 | 9.20 |
| Pubmed | $84.30_{\pm0.90}$ | $87.97_{\pm1.02}$ | $\underline{88.72}_{\pm1.19}$ | $\mathbf{89.95}_{\pm0.50}$ | $83.54_{\pm2.57}$ | -6.41 | -4.43 | -0.76 |
| CS | $\mathbf{97.99}_{\pm0.23}$ | $95.03_{\pm1.34}$ | $95.39_{\pm0.44}$ | $95.22_{\pm0.22}$ | $\underline{95.97}_{\pm0.34}$ | 0.58 | 0.94 | -2.02 |
| Physics | $\mathbf{98.84}_{\pm0.12}$ | $95.72_{\pm0.30}$ | $96.43_{\pm0.20}$ | $\underline{96.80}_{\pm0.24}$ | $94.78_{\pm0.34}$ | -2.02 | -0.94 | -4.06 |
| Computers | $\mathbf{98.07}_{\pm0.10}$ | $93.22_{\pm0.16}$ | $\underline{93.30}_{\pm0.32}$ | $92.96_{\pm0.39}$ | $92.09_{\pm0.73}$ | -1.21 | -1.13 | -5.98 |
| Photos | $\mathbf{98.35}_{\pm0.16}$ | $94.21_{\pm0.28}$ | $\underline{94.69}_{\pm0.09}$ | $93.79_{\pm0.43}$ | $92.09_{\pm0.59}$ | -2.60 | -2.12 | -6.26 |

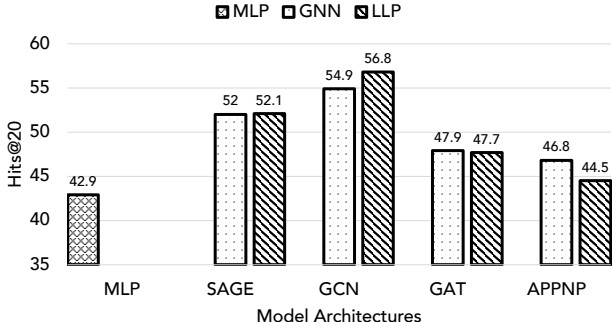

Figure 5: Link prediction performance measured by Hits@20 on Pubmed under the transductive setting with different GNN teachers (SAGE, GAT, GCN, and APPNP).

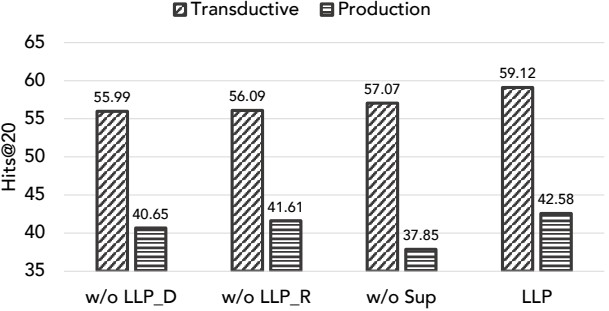

Figure 6: The averaged results of Hits@20 across all the datasets by dropping each component in Equation (7).

$\mathcal{L}_{\mathsf{LLP\_R}}$, $\mathcal{L}_{\mathsf{LLP\_D}}$, and $\mathcal{L}_{sup}$) is removed, which shows the importance of each component. It also demonstrates that $\mathcal{L}_{\mathsf{LLP\_R}}$ and $\mathcal{L}_{\mathsf{LLP\_D}}$ indeed provide complementary link prediction-related information for the student. Other than these two components, we find that the true link label information also contributes, especially under the production setting. In the production setting, as the neighbor information is sparse or absent, the limited true label information becomes critically important.

## D.5 Necessity Analysis of $\delta$ in $\mathcal{L}_{\mathsf{LLP\_R}}$

To analyze the necessity of $\delta$ in $\mathcal{L}_{\mathsf{LLP\_R}}$, we conduct the experiments on all non-OGB datasets to compare the results using $\mathcal{L}_{\mathsf{LLP\_R}}$ with and without $\delta$. The results are shown in Table 9. We observe that the results of $\mathcal{L}_{\mathsf{LLP\_R}}$ without $\delta$ always approach to zero after several training epochs. It demonstrates the effectiveness of $\delta$ in avoiding noise and transferring useful knowledge to the student.

## D.6 Comparison of task performance and inference time using different inference acceleration methods

In Table 10, we compare both the task performance and inference time using different acceleration methods on the two large-scale OGB datasets. From this table, we can observe that LLP shares

Table 9: The performance measured by Hits@20 of $\mathcal{L}_{\mathsf{LLP\_R}}$ with and without $\delta$.

| Method | Cora | Citeseer | Pubmed | CS | Physics | Computers | Photos |
|---|---|---|---|---|---|---|---|
| $\mathcal{L}_{\mathsf{LLP\_R}}$ | 76.52 | 75.23 | 53.30 | 68.30 | 60.28 | 25.98 | 33.33 |
| $\mathcal{L}_{\mathsf{LLP\_R}}$ w/o $\delta$ | 0.00 | 0.00 | 0.00 | 0.00 | 0.00 | 0.00 | 0.00 |

Table 10: Performance and inference time comparison of LLP with GNN inference acceleration methods. The performance is the averaged test result over 10 runs.

| Dataset | Metric | SAGE | QSAGE | PSAGE | Neighbor Sample | MLP | LLP |
|---------|--------|------|-------|-------|-----------------|-----|-----|
| OGB-Collab | Hits@50 | 48.69 | 45.36 | 48.34 | 31.50 | 36.95 | 45.27 |
| | Time (ms) | 134.3 | 128.3 | 128.7 | 28.6 | 1.9 | 1.9 |
| OGB-Citation2 | MRR | 82.56 | 82.53 | 82.04 | 79.82 | 40.63 | 53.20 |
| | Time (ms) | 2243.7 | 2206.4 | 2209.1 | 146.0 | 2.9 | 2.9 |

the same inference time with MLP, which is $15.12\times$ and $50.51\times$ faster than the most efficient acceleration method Neighbor Sample on OGB-Collab and OGB-Citation2, respectively. Our method outperforms MLP with large margins on both datasets. Although Neighbor Sample achieved certain speedup comparing to GNNs, and sometimes better prediction performance than LLP, it is still less competitive than LLP in production applications given the huge speed difference, which is critical for deployed models that require low latency.

Namely, our work considers KD strategies as a viable option to distill information from GNNs to MLPs owing to the large practical advantages that MLPs enjoy; these advantages (namely, latency) make them a leading choice for production systems compared to GNNs (Covington et al., 2016; Gholami et al., 2021; Zhang et al., 2021b), because GNNs suffer neighborhood explosion and data dependency downsides. Since violating tight latency constraints may not be feasible in production applications (e.g. "every 100ms of latency cost ... 1% in sales" [2], or "a 100-millisecond delay in website load time can hurt conversion rates by 7 percent" [3]), we may not be able to enjoy the strong performance of a GNN in such a setting (despite knowing it may perform better). With this perspective (that we must deploy a fast MLP model and cannot tolerate a slow GNN model), we can observe that our work proposes an effective KD strategy to significantly boost MLP performance with additional knowledge distilled from GNNs, offering significant advantages. Since in practice, it is common to operate under such constraints, the consistent improvements we show over MLP in all datasets (Table 1 and Table 2) has strong considerations for production settings which are MLP-constrained regardless of the performance gap with respect to GNN. That said, we hope that future work can ideally bridge the gap between LLP and GNN results in all settings.

### D.7   ANALYSIS OF THE RESULTS ON OGB-CITATION2

From Table 2, we can observe that LLP perform not well on OGB-Citation2 compared with other datasets. To understand the the reason for worse performance (size, or peculiarity of the dataset), we reduced the size of the original dataset by sampling a smaller graph from the original graph, and comparing the performance of the original and downsampled graphs. We conduct this experiments on two larger-scale datasets, OGB-Collab and OGB-Citation2. Based on the conclusion of Leskovec & Faloutsos (2006), that sampling based on random walks best-preserving certain properties of the original graph, we adopt this strategy to generate the downsampled graphs. For OGB-Collab, we sampled a graph with the similar size to Computers. For OGB-Citation2, we sampled a graph with the similar size like OGB-Collab. For these 4 datasets (original and downsampled versions of OGB-Collab and OGB-Citation2), we run GNN, MLP and LLP and report results in Table 11. This table yields 3 observations:

- The performance gap between GNN and LLP does not change significantly from the original graph to the downsampled graph on both datasets.

- The performance gap between GNN and LLP on OGB-Citation2 is always much larger than on OGB-Collab: 29.36 vs. 3.42 on the original graphs, and 10.76 vs. 2.07 on the downsampled graphs.

---

[2]https://www.gigaspaces.com/blog/amazon-found-every-100ms-of-latency-cost-them-1-in-sales
[3]https://www.prnewswire.com/news-releases/akamai-online-retail-performance-report-milliseconds-are-critical-300441498.html

Table 11: The statistics of the original and sampled graph of `OGB-Collab` and `OGB-Citation2` and the experimental results on these graphs.

|  | OGB-Collab | | OGB-Citation2 | |
|---|---|---|---|---|
|  | Original | Sampled | Original | Sampled |
| #Nodes | 235,868 | 23,891 | 2,927,963 | 122,440 |
| #Edges | 1,285,465 | 188,640 | 30,561,187 | 1,366,101 |
| GNN | 48.69 | 83.31 | 82.56 | 39.99 |
| MLP | 36.95 | 75.70 | 40.63 | 24.34 |
| LLP | 45.27 | 81.24 | 53.20 | 29.23 |

Table 12: The performance measured by Hits@20 of LLP and GNN+FA (Alon & Yahav, 2020) on the cold-start nodes.

| Method | Cora | Citeseer | Pubmed | CS | Physics | Computers | Photos |
|---|---|---|---|---|---|---|---|
| GNN | 6.39 | 11.04 | 4.63 | 9.46 | 5.46 | 1.53 | 0.87 |
| GNN+FA | 2.03 | 2.89 | OOM | OOM | OOM | OOM | OOM |
| LLP | 22.01 | 32.09 | 37.68 | 46.83 | 39.37 | 14.64 | 23.79 |

- Although the downsampled graph from `OGB-Citation2` is a similar size to the `OGB-Collab` original graph, the performance gap between GNN and LLP on the downsampled graph based on `OGB-Citation2` is still much larger than the gap on `OGB-Collab` original graph.

Summarily, point 1 demonstrates that LLP 's performance is not significantly influenced by the size of the datasets. Both point 2 and 3 indicate that LLP 's worse performance on `OGB-Citation2` is due to a peculiarity of this particular dataset, rather than its size.

## D.8    FURTHER COMPARISON RESULTS ON COLD START NODES.

We further compare LLP with another related work (Alon & Yahav, 2020), which does not only rely on the connection relationship in the graph either. This paper identifies a bottleneck of graph neural networks, and proposes to modify the last layer to be a fully-adjacent layer (FA) as a simple strategy to circumvent the bottleneck problem. To evaluate its performance on cold-start setting, we modify the last layer of GNNs with a fully-adjacent layer following it. The results compared with LLP are shown in Table 12. We observe that this method is not suitable for large datasets – it results in a dense $N$ adjacency matrix and results in each node receiving messages from $N$ other nodes (which is problematic when $N$ is large). Most datasets we utilized in our paper with a fully-adjacent layer can not fit into an NVIDIA A100 GPU (40GB memory). The performance of GNN+FA on `Cora` and `Citeseer` is even worse than vanilla GNNs. The potential reason is link prediction tasks do not heavily depend on long-range information, where the best results were found using only 2-3 layers - we observe that Alon & Yahav (2020) focuses evaluation on smaller datasets which are sensitive to long-range dependencies, which seems to be a mismatch with our intended setting.

## D.9    COMPARISON BETWEEN LLP AND RANKDISTIL (REDDI ET AL., 2021)

RankDistil (Reddi et al., 2021) is designed to transfer ranking knowledge from the teacher to student, where the ranking is their ranking task training signal generated by the teacher. In LLP, we consider a link prediction task on graph data, which is standardly approached as a binary classification task (predicting link existence vs non-existence), and not a ranking one. We proposed our KD framework to distill relational graph information from the teacher to the student. Different from RankDistil which distills the information in a non-graph context, we distill in a graph context to keep the graph structure information. The difference in motivation across the methods leads to different choices in sampling and matching. **For sampling method**, our method samples nodes which are utilized to teach the

Table 13: The performance measured by Hits@20 of LLP and RandDistil (Reddi et al., 2021).

|            | Cora  | Citeseer | Pubmed | CS    | Physics | Computers | Photos |
|------------|-------|----------|--------|-------|---------|-----------|--------|
| RankDistil | 74.29 | 70.44    | 39.28  | 44.55 | 49.11   | 15.64     | 28.75  |
| LLP        | 78.82 | 77.32    | 57.33  | 68.62 | 72.01   | 35.32     | 49.32  |

student based on the graph structure unlike RankDistil, which is very important for the link prediction on the graph data; RankDistil thereby struggles in effectively the student about this graph structure. Moreover, RankDistil samples nodes used to teach students based on the teacher's result, which differs from our method which samples independently of the teacher. The choice made by RankDistil here can potentially make it even harder for the student to learn graph structure in the case of teacher mis-predictions (since this also influences the sampling). **For matching method**, our method matches the order and the distribution of the sampled node pairs between teacher and student, which we show are both useful and complementary (Table 4) in preserving structure information distillation between the two. Different from our work, RankDistil keeps the order for "top-K" items and penalize high scores by the student for "bottom-K" items (since there are too many candidate items). However, compared with keeping the order, penalizing the large scores is a "weaker" alignment method, which may be not beneficial for preserving graph structure information.

In addition to our discussion of conceptual differences and potential implications above, we also conduct some experiments to evaluate the impact of these differences in our link prediction task, by comparing the performance of LLP to RandDistil. We applied RankDistil on our task, where we take all the other nodes existing in the graph as the candidate "items" for each anchor node (as they would be considered in the RankDistil setup), and use RankDistil's matching methods to align the ranking results generated by teacher and student. Due to the lack of guideline for the parameter settings for RankDistil, we conduct the hyperparameter search of the number of top items from [5, 10, 20, 50] and the number of bottom items from [10, 50, 100, 200]. The results are shown in Table 13. We observe that LLP consistently outperforms RankDistil on all the datasets. We believe this demonstrates that sampling the context nodes w.r.t the anchor node based on the graph topology structure is more effective in preserving relevant graph structure and link prediction-related knowledge than using all the other nodes in the graph to sample from a teacher-based ranking. Moreover, different matching methods we propose also help distill more task-relevant information than RankDistil in our setting.

## E  IMPLEMENTATION DETAILS

**Transductive Setting.** Inspired by GLNN (Zhang et al., 2021b), we enlarge the size of the student MLP in our experiment. As suggested by GLNN, this can significantly shorten the gap between the student MLP and the teacher GNN without greatly reducing the timing performance. We set the hidden dimension of student MLP two times larger than the teacher for Physics, Computers, and Photos, and set it four times larger than the teacher for OGB-Collab and OGB-Citation2. We examine the timing performance of the enlarged students by repeating the inference task ten times. The inference time of LLP increases from 1.9 to 7.1 seconds on OGB-Collab and from 2.9 to 15.2 seconds on OGB-Citation2, but it is still $18.9\times$ and $147\times$ faster than SAGE, respectively.

**Model Hyper-parameters.** We take 2-layer SAGE (hidden size is set to 256) as the teacher for all the non-OGB datasets. For OGB-Collab and OGB-Citation2, we follow their official implementation to set the layer size as 3. We take 3-layer MLP as the student on these two datasets. For LLP, we conduct the hyperparameter search of the weights for $\mathcal{L}_{sup}$, $\mathcal{L}_{\mathsf{LLP\_R}}$ and $\mathcal{L}_{\mathsf{LLP\_D}}$ from [0.001, 0.01, 0.1, 1, 10, 100, 1000], the number of the nearby nodes $p$ from [1,2,3,4,5], the random sampling rate $q/p$ from [1, 3, 5, 10, 15], the learning rate from [0.001, 0.005] and the dropout rate from [0, 0.5].

**Implementation and Hardware Details.** Our code is implemented based on PyTorch Geometric (Fey & Lenssen, 2019). We conduct our experiments with NVIDIA V100 GPU(16GB memory). For OGB-Citation2, we run the experiments on NVIDIA A100 GPU with 40GB memory.

