# OpenReview forum: "Linkless Link Prediction via Relational Distillation"
_ICLR.cc/2023/Conference — Submitted to ICLR 2023_

### Official Review · Reviewer_Z2kr · 2022-10-24

**Confidence:** 4
**Correctness:** 4
**Technical Novelty And Significance:** 2
**Empirical Novelty And Significance:** 3
**Recommendation:** 5

**Clarity, Quality, Novelty And Reproducibility:**

The presentation of the paper is clear and easy to follow, but the novelty is limited in terms of the technical contribution. See the above section on strengths and weaknesses.


**Strength And Weaknesses:**

Strengths

- The proposed KD strategy for link prediction is interesting and more effective than the previously proposed KD strategy for node-level and graph-level tasks.

- The proposed method shows a faster inference time and even better performance than GNNs.
- The proposed method can be used meaningfully in a real-world scenario by showing excellent performance in cold start nodes, which have relatively few connections.

Weaknesses

- A lack of technical contribution. As mentioned by the author, from the perspective that link prediction is considered as a ranking task, the proposed method for knowledge distillation (margin-based ranking loss and distribution-based matching loss) were already proposed in RankDistill [1] which proposes knowledge distillation for the ranking task.

- Knowledge Distillation becomes more significant as the scale of the graph increases due to the high latency of GNN, but the huge performance degradation in the largest scale graph (ogbl-citation2) raises questions about the necessity of KD. I wonder if it is a special case in ogbl-citation2 or if the performance drop of KD usually occurs in large-scale graphs.

[1] Reddi, Sashank, et al. "Rankdistil: Knowledge distillation for ranking." International Conference on Artificial Intelligence and Statistics. PMLR, 2021.

**Summary Of The Paper:**

This paper presents a knowledge distillation framework for link prediction, Linkless Link Prediction (LLP). Unlike simple knowledge distillation methods that match outputs or representations of models, LLP distills relational knowledge of each link to the student MLP using proposed rank-based and distribution-based matching. These two matching strategies enable the student model to learn relational knowledge about context nodes. Experiments conducted on 9 benchmarks show the effectiveness and efficiency of the proposed framework.

**Summary Of The Review:**

This paper presents a knowledge distillation framework for link prediction. The presentation of the paper is clear and easy to follow, and the motivation is clear, but the technical novelty is limited.

---

> ### Author Response · Authors · 2022-11-17
> **Response to Reviewer Z2kr [4/4]**
>
> Thanks again for your comments and diligence in reviewing our work. We hope our responses have addressed your concerns.  If so, we hope that you will consider raising your score. If there are any notable points unaddressed, please let us know and we will be happy to respond.
>
> Kind Regards,\
> LLP Authors

---

> ### Author Response · Authors · 2022-11-17
> **Response to Reviewer Z2kr [3/4]**
>
> >Q2. Knowledge Distillation becomes more significant as the scale of the graph increases due to the high latency of GNN, but the huge performance degradation in the largest scale graph (ogbl-citation2) raises questions about the necessity of KD. I wonder if it is a special case in ogbl-citation2 or if the performance drop of KD usually occurs in large-scale graphs.
>
> This is indeed a good question. To understand the the reason for worse performance (size, or peculiarity of the dataset), we reduced the size of the original dataset by sampling a smaller graph from the original graph, and comparing the performance of the original and downsampled graphs. We conduct this experiments on two larger-scale datasets, OGB-Collab and OGB-Citation2. Based on the conclusion of [2], that sampling based on random walks best-preserving certain properties of the original graph, we adopt this strategy to generate the downsampled graphs. For OGB-Collab, we sampled a graph with the similar size to Computers. For OGB-Citation2, we sampled a graph with the similar size like OGB-Collab.  For these 4 datasets (original and downsampled versions of OGB-Collab and OGB-Citation2), we evaluate GNN, MLP and LLP and report results in the below table:
>
> |        | OGB-Collab |         | OGB-Citation2 |           |
> |--------|:----------:|:-------:|:-------------:|:---------:|
> |        | Original   | Sampled | Original      | Sampled   |
> | #Nodes | 235,868    | 23,891  | 2,927,963     | 122,440   |
> | #Edges | 1,285,465  | 188,640 | 30,561,187    | 1,366,101 |
> | GNN    | 48.69      | 83.31   | 82.56         | 39.99     |
> | MLP    | 36.95      | 75.70   | 40.63         | 25.34     |
> | LLP    | 45.27      | 81.24   | 53.20         | 29.23     |
>
> This table yields 3 observations, which help answer your question:
>  1. The performance gap between GNN and LLP does not change significantly from the original graph to the downsampled graph on both datasets.
>  2. The performance gap between GNN and LLP on OGB-Citation2 is always much larger than on OGB-Collab: 29.36 vs. 3.42 on the original graphs, and 10.76 vs. 2.07 on the downsampled graphs.
>  3. Although the downsampled graph from OGB-Citation2 is a similar size to the OGB-Collab original graph, the performance gap between GNN and LLP on the downsampled graph based on OGB-Citation2 is still much larger than the gap on OGB-Collab original graph.
>
> Summarily, point 1 demonstrates that LLP's performance is not significantly influenced by the size of the datasets. Both point 2 and 3 indicate that LLP's worse performance on OGB-Citation2 is due to a peculiarity of this particular dataset, rather than its size.
>
> Thanks for the question, as the results help us strengthen our experiments. **We have also included this table in Appendix D.7.**
>
> Additionally, we want to clarify that our work considers KD strategies as a viable option to distill information from GNNs to MLPs owing to the large practical advantages that MLPs enjoy; these advantages (namely, latency) make them a leading choice for production systems compared to GNNs [3,4,5], because GNNs suffer neighborhood explosion and data dependency downsides.  Since violating tight latency constraints may not be feasible in production applications (e.g. "every 100ms of latency cost ... 1% in sales" [6], or "a 100-millisecond delay in website load time can hurt conversion rates by 7 percent" [7]), we may not be able to enjoy the strong performance of a GNN in such a setting (despite knowing it may perform better).  With this perspective (that we must deploy a fast MLP model and cannot tolerate a slow GNN model), we can observe that our work proposes an effective KD strategy to significantly boost MLP performance with additional knowledge distilled from GNNs, offering significant advantages. Since in practice, it is common to operate under such constraints, the consistent improvements we show over MLP in all datasets (Table 1 and Table 2) has strong considerations for production settings which are MLP-constrained regardless of the performance gap with respect to GNN. That said, we hope that future work can ideally bridge the gap between LLP and GNN results in all settings.
>
> [2] Leskovec et al., Sampling from large graphs. KDD 2006.\
> [3] Covington et al., Deep neural networks for youtube recommendations. RecSys 2016.\
> [4] Gholami et al., A survey of quantization methods for efficient neural network inference. arXiv 2021.\
> [5] Zhang et al., Graph-less neural networks: Teaching old mlps new tricks via distillation. arXiv 2021.\
> [6] https://www.gigaspaces.com/blog/amazon-found-every-100ms-of-latency-cost-them-1-in-sales \
> [7] https://www.prnewswire.com/news-releases/akamai-online-retail-performance-report-milliseconds-are-critical-300441498.html

---

> ### Author Response · Authors · 2022-11-17
> **Response to Reviewer Z2kr [2/4]**
>
> - More technically and specifically, our work adopts different sampling and matching methods.
>     * Sampling method.
>         * RankDistil asks the teacher to first rank all the items without sampling, and then sample the positive and negative pools based on its result. It further samples the "top-K" ranked items and "bottom-K" ranked items to teach the student.
>         * LLP aims to distill relational graph structure knowledge. So, our sampling method is to choose "context" nodes for each "anchor" node using the graph structure. As we introduced in the Practical Implementation of LLP in Section 3.3, we sample nearby nodes via fixed-length random walks to preserve the local structure and randomly sample some nodes from the whole graph to additionally preserve the global structure for each anchor node.
>         * **[Difference]** Our method samples nodes which are utilized to teach the student *based on the graph structure* unlike RankDistil, which is very important for the link prediction on the graph data; RankDistil thereby struggles in effectively the student about this graph structure. Moreover, RankDistil samples nodes used to teach students *based on the teacher’s result*, which differs from our method which samples independently of the teacher.  The choice made by RankDistil here can potentially make it even harder for the student to learn graph structure in the case of teacher mis-predictions (since this also influences the sampling).
>     * Matching method.
>         * RankDistil takes different matching methods for positive samples and negative samples. It matches the order of the "top-K" teacher ranking as the positives between teacher and student. Then, for the negatives, it penalizes the student for producing large scores for these items.
>         * LLP samples some context nodes for each anchor node using the graph structure. We match the order and the distribution of all the context nodes w.r.t the anchor nodes.
>         * **[Difference]** Our method matches the order and the distribution of the sampled node pairs between teacher and student, which we show are both useful and complementary (Table 4) in preserving structure information distillation between the two. Different from our work, RankDistil keeps the order for "top-K" items and penalize high scores by the student for "bottom-K" items (since there are too many candidate items). However, compared with keeping the order, penalizing the large scores is a "weaker" alignment method, which may be not beneficial for preserving graph structure information.
>
> In addition to our discussion of conceptual differences and potential implications above, we also conduct some experiments to evaluate the impact of these differences in our link prediction task, by comparing the performance of LLP to RankDistil. We applied RankDistil on our task, where we take all the other nodes existing in the graph as the candidate "items" for each anchor node (as they would be considered in the RankDistil setup), and use RankDistil's matching methods to align the ranking results generated by teacher and student. Due to the lack of guideline for the parameter settings for RankDistil, we conduct the hyperparameter search of the number of top items from [5, 10, 20, 50] and the number of bottom items from [10, 50, 100, 200]. The results are shown below.
>
> |             | Cora   | Citeseer | Pubmed | CS    | Physics | Computers | Photos |
> |-------------|--------|----------|--------|-------|---------|-----------|--------|
> | RankDistil  | 74.29  | 70.44    | 39.28  | 44.55 | 49.11   | 15.64     | 28.75  |
> | LLP         | **78.82**  | **77.32**    | **57.33**  | **68.62** | **72.01**   | **35.32**     | **49.32** |
>
> We observe that LLP consistently outperforms RankDistil on all the datasets. We believe this demonstrates that sampling the context nodes w.r.t the anchor node based on the graph topology structure is more effective in preserving relevant graph structure and link prediction-related knowledge than using all the other nodes in the graph to sample from a teacher-based ranking. Moreover, the different matching methods we propose also help distill  more task-relevant information than RankDistil in our setting. **We have added RankDistil in our related work (Appendix A) and the experimental results in Appendix D.9.**  Thanks again for pointing us to this work -- including these experiments strengthens our work.
>
> [1] Reddi et al. "RankDistil: Knowledge distillation for ranking." AISTATS, 2021

---

> ### Author Response · Authors · 2022-11-17
> **Response to Reviewer Z2kr [1/4]**
>
> Dear Reviewer Z2kr:
>
> Thanks for your comments and insightful questions. We hope our following point-to-point responses can address your concerns. (Sorry for splitting the responses into multiple parts due to the limits on the character number.)
>
> >Q1. A lack of technical contribution given the existence of RankDistil [1]
>
> Thanks for pointing us to this related work; while RankDistil [1] certainly appears to be an interesting and impactful work on knowledge distillation for ranking, we do not believe its existence mitigates or prevents our technical contributions, given the following reasons:
> - Our proposal of knowledge distillation has different motivations:
>     * RankDistil is designed to **transfer ranking knowledge** from the teacher to student, where the ranking is their ranking task training signal generated by the teacher.  This makes it somewhat similar to logit-based matching KD in its intent (matching raw ranking scores assigned to each item). But, given the large item space, matching all  orders generated by teacher and student may involve too much noise. Hence, RankDistil proposes using the teacher to rank all the items, and asking the student to preserve the "top-K" teacher ranking, and penalizing the student for high scores on the "bottom-K" teacher ranking.
>     * In LLP, we consider a link prediction task on graph data, which is standardly approached as a binary classification task (predicting link existence vs non-existence), and not a ranking one. As such, we start with logit-based matching KD (Section 3.1), where we match node-pair scores generated by the teacher and student. We also further evaluate representation based matching KD. However, we find that both of these approaches do not work well for our task. We hypothesize that the reason for the failure of these two KD methods is that link prediction on the graph data is heavily dependent on the **relational, graph structure** information (Section 3.2). Based on this hypothesis, we propose a KD framework to **distill relational graph information** from the teacher to the student, thereby adding a ranking interpretation (Section 3.3). To achieve this target, we sample context nodes with respect to each anchor node. We match the distribution or the rank generated of all the context nodes w.r.t the anchor node to retain task-relevant graph knowledge.
>     * **[Difference]** Summarily, the difference in motivation across the methods leads to different choices in sampling and matching, motivated by the underlying differences of distilling ranking as RankDistil does (in a non-graph context) between teacher and student, versus distilling relational information as we do (in a graph context).

---

> ### Author Response · Authors · 2022-12-07
> **Looking forward to your response**
>
> Dear Reviewer Z2kr,
>
> We sincerely appreciate your valuable comments. We took significant efforts during the rebuttal to clarify misunderstandings and address your concerns with our responses.
>
> Since the discussion period is close to the end, could you kindly look at our responses and raise your score if you believe we have satisfactorily addressed your concerns? If you have any further concerns, we will be happy to respond.
>
> Best Regards,\
> LLP Authors

---

> ### Author Response · Authors · 2022-12-12
> **Last day reminder**
>
> Dear Reviewer Z2kr,
>
> Thank you again for your review! Today is the last day for the discussion. We do hope you can take a look at our responses to your comments and give us some feedback. We believe we have addressed all your questions and concerns in this revision. If so, please consider raising your score. If there are any further concerns, please let us know, and we will gladly address them.
>
> Thanks,\
> LLP Authors

---

### Official Review · Reviewer_4BGr · 2022-10-24

**Confidence:** 4
**Correctness:** 3
**Technical Novelty And Significance:** 2
**Empirical Novelty And Significance:** 3
**Recommendation:** 8

**Clarity, Quality, Novelty And Reproducibility:**

Clarity:
 - Very good

Reproducibility:
 - Excellent

Quality:
 - Good

Novelty:
 - Medium

**Strength And Weaknesses:**

Strengths:
 - The paper is well written and clear.
 - The experiments and models are described in a reproducible manner.
 - The proposed production setting and analysis of cold-start nodes is interesting and valuable to recommender systems.
 - The proposed method seems sensible.

Weaknesses:
 - The method is significantly outperformed on the only large link prediction graph provided (OGB-Citation2 Tab.1). This is problematic because the purpose of the method is to work on large-scale graphs, where GNNs might be too expensive to run.
 - The comparison against GNN methods on cold-start nodes is not fair as known GNN techniques exist that might help in this setting (see [1], where a fully adjacent layer is proposed).
 - The evaluation method leaves me with some doubt as I think it is biased (if my understanding is correct) due to no ranking against all nodes in the graph (as is standard in KG link prediction). This is an unrealistic test setting that is also biased.
 - Knowledge graph completion is an important link prediction task that deserves to be mentioned in the introduction and related work, with a brief explanation why this setting was ignored.
 - The GNN scaling method comparison is a bit weak in that some key methods have not been considered as well as only considering inference time, but not performance. There will always be a trade-off. In general, the paper is too quick to assume that knowledge distillation to MLPs is needed and cannot be dealt with by accelerating GNNs, a point that needs further evidence before it can be accepted.

[1] https://arxiv.org/pdf/2006.05205.pdf

**Summary Of The Paper:**

The paper proposes to distill GNN knowledge for link prediction into MLPs via distribution and rank based losses as opposed to prior work which only uses logit-matching or representation matching. The authors test their method against these prior knowledge distillation methods as well as stand-alone MLPs and the original teacher GNN. The authors further observe the difficulty with regards to cold-start nodes and observe significant performance improvements compared to some scaling GNN methods.

**Summary Of The Review:**

The paper has excellent scientific communication and reproducibility. I want to highlight this as I believe papers in the field should be held to higher standards in this regard. So I would like to commend the authors on these points. The method is well motivated, but not technically sophisticated.

In my mind there is three barriers to acceptance:
1. I understand that there is a lack of large-scale homogeneous link prediction datasets out there. However, given the purpose of the knowledge distillation is to be applied on large scale graphs the performance in Table 1 on OGB-Citation2 is problematic. For acceptance, I would need to see a convincing analysis that shows that this is not due to the size of the dataset (the method works well on smaller citation graphs,e.g. Citeseer), but rather a peculariaty of this particular dataset. For instance, currently it is not unreasonable to hypothesis that MLPs will struggle to compress the information of many node comparisons efficiently, thus harming performance as the training graph gets larger, which would be a fundamental problem.
2. The argument that MLPs are better suited to cold-start nodes is interesting and gives this direction additional value. However, currently I  believe the comparison in table 3, needs adaption to be convincing. There are several GNN architectures that have proposed that do not only rely on the connectivity of the graph. The simplest example is [1], where a fully adjacent layer is used at the end, if the MLP can still out-perform such a GNN architecture, then I am willing to accept that KD into MLPs might be key for recommender systems, where cold-start nodes will indeed be common.
3. The comparison against GNN scaling methods needs a bit of improvement, namely I would like to see a discussion of the trade-off of inference time on the large-scale datasets (such as OGB-Citation2) and the final performance (as we can always predict faster while doing so less accurately). As well as including some more recent methods such as [2].

I appreciate that this is a lot of work for a rebuttal phase, but otherwise the paper remains too unconvincing for such a top-level conference. Should the authors be able to address the three points above, I am willing to raise my score to acceptance of the paper.

EDIT: The authors rebuttal has addressed my concerns. I have thus raised my score for the paper to be accepted.

[1] https://arxiv.org/pdf/2006.05205.pdf
[2] http://proceedings.mlr.press/v139/fey21a/fey21a.pdf

---

> ### Author Response · Authors · 2022-11-17
> **Response to Reviewer 4BGr [3/3]**
>
> >Q4. The evaluation method leaves me with some doubt as I think it is biased (if my understanding is correct) due to no ranking against all nodes in the graph (as is standard in KG link prediction). This is an unrealistic test setting that is also biased.
>
> We agree that the ranking against all nodes in the graph is a reasonable and effective evaluation method for link prediction, which is standard in KG link prediction. However, this process leads to high time cost on large-scale link prediction tasks like the datasets provided on OGB [9]. To make the evaluation metrics for each dataset reasonable, OGB provides official splitting methods and evaluation metrics for each dataset according to its different context, based on established protocols in relevant literature. On undirected graphs, OGB evaluates the performance by ranking each positive edge in the validation/test set against random-sampled "negative" edges, which is aimed at evaluating how well the model ranks performs in a plausible recommendation context. Researchers in (homogeneous) link prediction literature commonly follow this evaluation setting [10,11,12]. For consistency and convention, we use Hits@20 on all undirected non-OGB datasets following this existing work [10,11,12].
>
> [9] https://ogb.stanford.edu/docs/leader_linkprop/ \
> [10] Yun et al., Neo-gnns: Neighborhood overlap-aware graph neural networks for link prediction. NeurIPS 2021.\
> [11] Zhang et al., Labeling trick: A theory of using graph neural networks for multi-node representation learning. NeurIPS 2021.\
> [12] Zhao et al., Learning from counterfactual links for link prediction. ICML 2022
>
> >Q5. Knowledge graph completion is an important link prediction task that deserves to be mentioned in the introduction and related work, with a brief explanation why this setting was ignored.
>
> Thanks for pointing out this task. **We have mentioned this task in our introduction and also added knowledge graph completion related work in our Further Related Work section (Appendix A).** We highlight the text in red in the updated PDF and also add a snippet below:
> “One line of work is the strategy we discussed in Section 2, where the GNN-based encoder learns node representations and the decoder predicts whether the link exists. It is worth mentioning that knowledge graph completion follows this strategy to predict not only the link existence but also the type of the link [13,14,15,16]. These methods mainly use heterogeneous graph neural networks sensitive to different edge types.”
>
> LLP did not include knowledge graph completion as the downstream tasks, because our work is exploratory and therefore focuses on a more basic, homogeneous graph setting. Our goal in this paper is to lay the groundwork for the direction of cross-model relational distillation of graph topology between GNNs and MLPs for link prediction tasks. With that said, extensions of LLP to heterogeneous graphs are certainly interesting and valuable to explore, especially for KG literature, and we feel this is a great opportunity for future work.
>
> [13] Schlichtkrull et al., Modeling relational data with graph convolutional networks. ESWC 2018.\
> [14] Nathani et al., Learning attention-based embeddings for relation prediction in knowledge graphs. arXiv 2019.\
> [15] Vashishth et al., Composition-based Multi-Relational Graph Convolutional Networks. ICLR 2020.\
> [16] Zhang et al., Few-shot knowledge graph completion. AAAI 2020.
>
> Thank you again for your comments and detailed review of our work. We hope that we have addressed your concerns, and that you will consider raising your score. If we have left any notable points of concern unaddressed, please let us know, and we will be happy to respond.
>
> Kind Regards,\
> LLP Authors

---

> ### Author Response · Authors · 2022-11-17
> **Response to Reviewer 4BGr [2/3]**
>
> >Q2. Whether LLP can outperform "+FA"[7] on cold-start settings.
>
> Thanks for pointing out this reference. This paper identifies a bottleneck of graph neural networks, and proposes to modify the last layer to be a fully-adjacent layer (FA) as a simple strategy to circumvent the bottleneck problem. It is a very interesting work, which demonstrably improves performance on some long-range dependent tasks. **To make our related work (Appendix A) comprehensive, we added a reference and discussed in our paper.**
>
> To evaluate the performance of [7] on cold-start setting, we modify the last layer of GNNs with a fully-adjacent layer following [7]. The results compared with LLP are shown as follows:
>
> |        | Cora  | Citeseer | Pubmed | CS    | Physics | Computers | Photos |
> |:------:|:-----:|:--------:|:------:|:-----:|:-------:|:---------:|:------:|
> | GNN    | 6.39  | 11.04    | 4.63   | 9.46  | 5.46    | 1.53      | 0.87   |
> | GNN+FA | 2.03  | 2.89     | OOM    | OOM   | OOM     | OOM       | OOM    |
> | LLP    | **22.01** | **32.09**    | **37.68**  | **46.83** | **39.37**   | **14.64**     | **23.79**  |
>
> We observe that this method is not suitable for large datasets -- it results in a dense $N \times N$ adjacency matrix and results in each node receiving messages from $N$ other nodes (which is problematic when $N$ is large). Most datasets we utilized in our paper with a fully-adjacent layer can not fit into an NVIDIA A100 GPU (40GB memory). The performance of GNN+FA on Cora and Citeseer is even worse than vanilla GNNs. The potential reason is link prediction tasks do not heavily depend on long-range information, where the best results were found using only 2-3 layers -- we observe that [7] focuses evaluation on smaller datasets which are sensitive to long-range dependencies, which seems to be a mismatch with our intended setting. Thanks for suggesting this experiment -- it strengthens our work. **We also added this comparison in Appendix D.8.**
>
> [7] Alon et al., On the bottleneck of graph neural networks and its practical implications. ICLR 2021.
>
> >Q3. The trade-off of inference time on the large-scale datasets and the final performance. Including more recent methods [8]
>
> Thanks for the suggestion. In the following table (also added as Table 10 in the revised paper) we compare both the task performance and inference time using different acceleration methods on the two large-scale OGB datasets:
>
> |               |          | SAGE   | QSAGE  | PSAGE  | Neighbor Sample | MLP   | LLP   |
> |:-------------:|:--------:|:------:|:------:|:------:|:---------------:|:-----:|:-----:|
> | OGB-Collab        | Hits@50  | **48.69**  | 45.36  | 48.34  | 31.50           | 36.95 | 45.27 |
> |               | Time(ms) | 134.3  | 128.3  | 128.7  | 28.6            | **1.9**   | **1.9**   |
> | OGB-Citation2 | MRR      | **82.56**  | 82.53  | 82.04  | 79.82           | 40.63 | 53.20 |
> |               | Time(ms) | 2243.7 | 2206.4 | 2209.1 | 146.0           | **2.9**   | **2.9**   |
>
> From this table, we can observe that LLP shares the same inference time with MLP, which is **15.12x** and **50.51x faster** than the most efficient acceleration method Neighbor Sample on OGB-Collab and OGB-Citation2, respectively. Our method outperforms MLP with large margins on both datasets. Although Neighbor Sample achieved certain speedup comparing to GNNs, and sometimes better prediction performance than LLP, it is still less competitive than LLP in production applications given the huge speed difference, which is critical for deployed models that require low latency (as we discuss more extensively in our response to **Q1** above).  **We have added this analysis in the Appendix D.6.**
>
> GNNAutoScale [8] is proposed to accelerate the training process of GNNs -- it samples mini-batches of nodes and retains the nodes inside the current mini-batch and their 1-hop neighbors. For the nodes outside of the mini-batch, it directly uses offline-stored historical embeddings. During inferencing, GNNAutoScale directly uses the historical embeddings of the last layer for all the nodes (assuming precomputation). However, in this case, the inference time of all the methods in the reported table can be reduced to a constant complexity by pre-computing and storing the node embeddings offline. Moreover, under the production setting where new nodes would appear after the training, GNNAutoScale would not have any cached historical embedding for the new nodes, and hence its inference time would be the same as other GNN-based methods. Therefore, we do not include GNNAutoScale as a baseline here. Nonetheless, it is indeed a relevant work on GNN acceleration and scalability, and **we included the relevant discussion in Further Related Work (Appendix A).**
>
> [8] Fey et al., Gnnautoscale: Scalable and expressive graph neural networks via historical embeddings. ICML 2021.

---

> ### Author Response · Authors · 2022-11-17
> **Response to Reviewer 4BGr [1/3]**
>
> Dear Reviewer 4BGr,
>
> Thank you for providing insightful and constructive comments for us. We hope our point-to-point responses can address your concerns. Please find our detailed response below. (Sorry for splitting the responses into multiple parts due to the limits on the character number.)
>
> >Q1. Performance in Table 1 on OGB-Citation2 is problematic... Convincing analysis that shows that this is not due to the size of the datasets but rather a peculiarity of this particular dataset
>
> Thanks for understanding that there is a lack of large-scale homogeneous link prediction datasets. This is indeed a good question. To understand the the reason for worse performance (size, or peculiarity of the dataset), we reduced the size of the original dataset by sampling a smaller graph from the original graph, and comparing the performance of the original and downsampled graphs. We conduct this experiments on two larger-scale datasets, OGB-Collab and OGB-Citation2. Based on the conclusion of [1], that sampling based on random walks best-preserving certain properties of the original graph, we adopt this strategy to generate the downsampled graphs. For OGB-Collab, we sampled a graph with the similar size to Computers. For OGB-Citation2, we sampled a graph with the similar size like OGB-Collab.  For these 4 datasets (original and downsampled versions of OGB-Collab and OGB-Citation2), we run GNN, MLP and LLP and report results in the below table:
>
>
> |        | OGB-Collab |         | OGB-Citation2 |           |
> |:------:|:----------:|:-------:|:-------------:|:---------:|
> |        | Original   | Sampled | Original      | Sampled   |
> | #Nodes | 235,868    | 23,891  | 2,927,963     | 122,440   |
> | #Edges | 1,285,465  | 188,640 | 30,561,187    | 1,366,101 |
> | GNN    | 48.69      | 83.31   | 82.56         | 39.99     |
> | MLP    | 36.95      | 75.70   | 40.63         | 25.34     |
> | LLP    | 45.27      | 81.24   | 53.20         | 29.23     |
>
>
> This table yields 3 observations, which help answer your question:
>  1. The performance gap between GNN and LLP does not change significantly from the original graph to the downsampled graph on both datasets.
>  2. The performance gap between GNN and LLP on OGB-Citation2 is always much larger than on OGB-Collab: 29.36 vs. 3.42 on the original graphs, and 10.76 vs. 2.07 on the downsampled graphs.
>  3. Although the downsampled graph from OGB-Citation2 is a similar size to the OGB-Collab original graph, the performance gap between GNN and LLP on the downsampled graph based on OGB-Citation2 is still much larger than the gap on OGB-Collab original graph.
>
> Summarily, point 1 demonstrates that LLP's performance is not significantly influenced by the size of the datasets. Both point 2 and 3 indicate that LLP's worse performance on OGB-Citation2 is due to a peculiarity of this particular dataset, rather than its size.
>
> Thanks for the question, as the results help us strengthen our experiments. **We have also included this table in Appendix D.7.**
>
> Additionally, we want to clarify that our work considers KD strategies as a viable option to distill information from GNNs to MLPs owing to the large practical advantages that MLPs enjoy; these advantages (namely, latency) make them a leading choice for production systems compared to GNNs [2,3,4], because GNNs suffer neighborhood explosion and data dependency downsides.  Since violating tight latency constraints may not be feasible in production applications (e.g. "every 100ms of latency cost ... 1% in sales" [5], or "a 100-millisecond delay in website load time can hurt conversion rates by 7 percent" [6]), we may not be able to enjoy the strong performance of a GNN in such a setting (despite knowing it may perform better).  With this perspective (that we must deploy a fast MLP model and cannot tolerate a slow GNN model), we can observe that our work proposes an effective KD strategy to significantly boost MLP performance with additional knowledge distilled from GNNs, offering significant advantages.  Since in practice, it is common to operate under such constraints, the consistent improvements we show over MLP in all datasets (Table 1 and Table 2) has strong considerations for production settings which are MLP-constrained regardless of the performance gap with respect to GNN. That said, we hope that future work can ideally bridge the gap between LLP and GNN results in all settings.
>
> [1] Leskovec et al., Sampling from large graphs. KDD 2006.\
> [2] Covington et al., Deep neural networks for youtube recommendations. RecSys 2016.\
> [3] Gholami et al., A survey of quantization methods for efficient neural network inference. arXiv 2021.\
> [4] Zhang et al., Graph-less neural networks: Teaching old mlps new tricks via distillation. arXiv 2021.\
> [5] https://www.gigaspaces.com/blog/amazon-found-every-100ms-of-latency-cost-them-1-in-sales \
> [6] https://www.prnewswire.com/news-releases/akamai-online-retail-performance-report-milliseconds-are-critical-300441498.html

---

> ### Author Response · Authors · 2022-12-04
> **Thank you**
>
> We truly appreciate your positive assessment after the rebuttal and your effort in helping us to strengthen the paper!
>
> Kind regards,\
> LLP Authors

---

### Official Review · Reviewer_Wrjf · 2022-11-01

**Confidence:** 3
**Correctness:** 3
**Technical Novelty And Significance:** 2
**Empirical Novelty And Significance:** 2
**Recommendation:** 5

**Clarity, Quality, Novelty And Reproducibility:**

Clarity: The paper is clearly written except some parts (see below). The paper seems to apply KD method directly into link prediction problem without a very clear motivation.

Quality: The paper is technically correct. Experiments are well conducted. I did not find any flaw.

Novelty: As mentioned before,  the paper plugs KD into a new task, which seems a new idea--- but I am not sure about the motivation

Reproducibility: No readme file present in the supplementary material.


**Strength And Weaknesses:**

Strengths:

+ Good writing
+ New application of knowledge distillation

Weaknesses

- Some parts of the experiments are not clear
- A clear motivation is lacking
- No readme file in the code--- this seriously constrains the reproducibility


**Summary Of The Paper:**

The paper aims to construct an architecture for Knowledge distillation from GNN based methology to MLP for link Prediction task in specific.
As GNNs are more powerful and more accurate than vanilla MLP for link Prediction task primarily due to their ability to aggregate neighbot good information, these results in more model complexity and computational cost. Hence GNNs for link Prediction tasks can't be deployed directly in a production setting. Whereas MLPs had relatively less computational cost and complexity but don't take into consideration neighborhood information which is curcial for link Prediction tasks. Hence the authors propose a knowledge distillation architecture from teacher GNN to student MLP which utilises the advantages of accuracy of GNNs as well as less model complexity of MLPs for a production setting.


**Summary Of The Review:**

Strengths

+ Motivation:  The authors have clearly outlined the motivation setting for such a proposal behind knowledge distillation.

+ Ablation studies confirm that their proposed model outperforms and close to GNN.

+ The have showcases on 2 large scale OG datasets, thereby confirming the effectiveness of their proposed model in large graphs.

+ The authors have  explains well the experimental setup in appendix. Well written.



Weakness/questions for authors

 - Not clear in experiments what LLMatch metric is used currently.

- Uses logit based matching based on a very old paper. And also doesn't showcase the motivation of proposal of logit based matching. It should be clearly substituted with recent paper evidences.

- The authors have performed experiments in large OGB datasets which are typically used for link Prediction settings. But as the authors have mentioned in the start of the paper for applications in an industrial setting, did they also perform experiments in a realtime production setting like a recommendation system e.t.c. That would have been nice to have. An analysis on a large scale real production environment(rather than simulation) would tell much more about the utility of proposed method. If there are limitations, that should be clearly specified.

- Inference time graphs are plotted, however very less conclusive analysis done on much of a significant time change does KD provides over GNN w.r.t accuracy.

- Each of the experiment tables should also have clear notation what values they are showing. Is it showing HITs of AUC. That should specified under each table. Some tables don't have titles or caption ( example ablation study)

---

> ### Author Response · Authors · 2022-11-17
> **Response to Reviewer Wrjf [4/4]**
>
> [14] Zhang et al., Link prediction based on graph neural networks. NeurIPS 2018.\
> [15] Yun et al., Neo-gnns: Neighborhood overlap-aware graph neural networks for link prediction. NeurIPS 2021.
>
> >Q7. No readme file in code package:
>
> Thanks for pointing this out. We have added a Readme file with detailed instructions and examples for running our code in the supplementary files. To further improve the reproducibility, we also cleaned our code and added comments for better readability.
>
> Thanks again for your valuable suggestions and comments which help us strengthen our paper.  We hope that we have addressed your concerns, and that you will consider raising your score. If we have left any notable points of concern unaddressed, please let us know, and we will be happy to respond.
>
> Kind regards,\
> LLP Authors

---

> ### Author Response · Authors · 2022-11-17
> **Response to Reviewer Wrjf [3/4]**
>
> Namely, our work considers KD strategies as a viable option to distill information from GNNs to MLPs owing to the large practical advantages that MLPs enjoy; these advantages (namely, latency) make them a leading choice for production systems compared to GNNs [12,13,14], because GNNs suffer neighborhood explosion and data dependency downsides.  Since violating tight latency constraints may not be feasible in production applications (e.g. "every 100ms of latency cost ... 1% in sales" [15], or "a 100-millisecond delay in website load time can hurt conversion rates by 7 percent" [16]), we may not be able to enjoy the strong performance of a GNN in such a setting (despite knowing it may perform better).  With this perspective (that we must deploy a fast MLP model and cannot tolerate a slow GNN model), we can observe that our work proposes an effective KD strategy to significantly boost MLP performance with additional knowledge distilled from GNNs, offering significant advantages.  Since in practice, it is common to operate under such constraints, the consistent improvements we show over MLP in all datasets (Table 1 and Table 2) has strong considerations for production settings which are MLP-constrained regardless of the performance gap with respect to GNN. That said, we hope that future work can ideally bridge the gap between LLP and GNN results in all settings.
>
> [12] Covington et al., Deep neural networks for youtube recommendations. RecSys 2016.\
> [13] Gholami et al., A survey of quantization methods for efficient neural network inference. arXiv 2021.\
> [14] Zhang et al., Graph-less neural networks: Teaching old mlps new tricks via distillation. arXiv 2021.\
> [15] https://www.gigaspaces.com/blog/amazon-found-every-100ms-of-latency-cost-them-1-in-sales \
> [16] https://www.prnewswire.com/news-releases/akamai-online-retail-performance-report-milliseconds-are-critical-300441498.html
>
> >Q5. Each of the experiment tables should also have clear notation
>
> Thanks for the suggestion. We added the evaluation metric information in the captions of all tables, and polished the captions to be more informative.
>
> >Q6. Novelty: As mentioned before, the paper plugs KD into a new task, which seems a new idea --- but I am not sure about the motivation
>
> **Motivation**: as discussed in the Introduction section, our work aims to take advantage of both GNN’s strong task performance and MLP’s huge speed advantages on link prediction tasks. Link prediction is a critical problem on graph data, and is a cornerstone of recommendation systems. Although GNNs have shown strong performance in link prediction, their success is heavily dependent on neighborhood fetching and aggregation schemes, which can lead to high time cost in training and inference compared to MLPs, especially on large-scale datasets. Compared with GNNs, MLPs usually have worse prediction performance but lower time cost in both training and inference, since they do not utilize graph topology information. Given these speed-performance tradeoffs, we proposed an innovative and effective KD framework to transfer knowledge from GNNs to MLPs to preserve strong link prediction performance while enjoying huge speed ups during inference.
>
> **Novelty**: we appreciate that you agree with our novelty about exploring KD in the contxt of a new task. We also want to take this oppotunity to further clarify the novelty of our work, which is not simply *plugging KD into a new task*. In short, our work proposed a new relational KD framework for link prediction. More specifically, our work can summarized into several steps:
> - We start with exploring two direct KD approaches (Section 3.1), namely logit-based matching and representation-based matching, and we show that they are not powerful enough to distill sufficient knowledge from teacher GNN to student MLP for strong link prediction performance.
> - We then hypothesize that the direct KD's poor performance is caused by their inability to capture the relational information on graph structure (Section 3.2), which are often heavily relied on by link prediction methods [3,14,15].
> - In accordance with our intuition regarding preservation of relational knowledge, we propose a **novel relational distillation framework** for link prediction, called LLP. It utilizes our proposed rank-based matching and distribution-base matching to distill the relational knowledge centered at each anchor node (details in Section 3.3).  This is a new, methodological contribution.
> - The comprehensive empirical study (Section 4) on 9 public benchmarks under three different settings demonstrates the effectiveness (consistently outperforming MLP with very large margins) as well as the efficiency (e.g., **776x** speed comparing to GNN on the large-scale OGB-Citation2 dataset) of our proposed method.

---

> ### Author Response · Authors · 2022-11-17
> **Response to Reviewer Wrjf [2/4]**
>
> >Q3. Should perform experiments in a realtime production setting.
>
> Thanks for your suggestion. We also agree that it would be ideal to show the effectiveness of our proposed method in a real production environment. However, building a production-grade solution is a costly investment in terms of engineers, time, and money. Moreover, we believe that our research effort and empirical evaluation in this work suggest that these investments could be worthwhile to implement in practice (subject to resourcing, priorities and strategic alignment in an industrial environment). Since these are quite challenging to tackle in the context of a research effort, we believe that a production-scale implementation may be out of the scope of our research project given the novelty and recency of our observations. With that said, we evaluate our methods as best possible:
> * We conduct our experiments using **nine benchmark datasets with various scales** from thousands to millions of nodes. We boost the performance of MLP with significant margins on **all datasets**, and even outperforms the teacher GNNs on **6 out of 9** benchmarks. Moreover, for the OGB-Citation2 dataset, we follow the official evaluation setting that ranks the candidate nodes for each target node, which resembles a realistic recommendation system setting where the candidate items are ranked for each target user.
> * We design a more realistic setting (called **production setting** in our paper) that mimics realistic use-cases for link prediction, where new nodes or edges appear frequently after the model training. Under this setting, our method obtains average performance improvemence of **12.01** points across all the datasets (each with its own metric as specified in the above answer to **Q1**).
> * We further conduct experiments on **cold start nodes**, which are  newly appeared nodes without any edges. Cold start recommendation is a very common challenge in production environments. Our experimental results show that LLP consistently outperforms GNN and MLP by an average of **25.29** and **9.42** points on Hits@20, respectively.
>
> As also mentioned in the Introduction (Section 1), GNNs will lead to high time cost during inference compared to MLP because the success of GNNs is heavily dependent on neighborhood fetching and aggregation schemes. This scenario will be made worse on larger-scale datasets, which is worth addressing. Thus, we propose this work to replace GNNs with MLP to reduce the inference time with limited performance sacrifice. As shown in Figure 2, we achieve an over **$700\times$** speed up compared to GNNs on the large-scale OGB-Citation2 dataset.
>
> We believe that the above results are sufficient to show that our method can reduce the inference time with significant performance implications in practical settings. We totally agree with your suggestion that it is worth also considering deploying LLP; we defer this setting to future work.
>
> >Q4. Inference time graphs are plotted, however very less conclusive analysis done on much of a significant time change does KD provide over GNN w.r.t accuracy.
>
> Thanks for the suggestion. In the following table (also added as Table 10 in the revised paper) we compare both the task performance and inference time using different acceleration methods on the two large-scale OGB datasets:
>
> |   |   | SAGE   | QSAGE  | PSAGE  | Neighbor Sample | MLP   | LLP   |
> |---------------|:--------:|:------:|:------:|:------:|:---------------:|:-----:|:----:|
> | OGB-Collab        | Hits@50  | **48.69**  | 45.36  | 48.34  | 31.50           | 36.95 | 45.27 |
> |     | Time(ms) | 134.3  | 128.3  | 128.7  | 28.6            | **1.9**   | **1.9**   |
> | OGB-Citation2 | MRR      | **82.56**  | 82.53  | 82.04  | 79.82           | 40.63 | 53.20 |
> |    | Time(ms) | 2243.7 | 2206.4 | 2209.1 | 146.0           | **2.9**   | **2.9**   |
>
> From this table, we can observe that LLP shares the same inference time with MLP, which is **15.12x** and **50.51x faster** than the most efficient acceleration method Neighbor Sample on OGB-Collab and OGB-Citation2, respectively. Our method outperforms MLP with large margins on both datasets. Although Neighbor Sample achieved certain speedup comparing to GNNs, and sometimes better prediction performance than LLP, it is still less competitive than LLP in production applications given the huge speed difference, which is critical for deployed models that require low latency.

---

> ### Author Response · Authors · 2022-11-17
> **Response to Reviewer Wrjf [1/4]**
>
> Dear Reviewer Wrjf,
>
> Thank you for your detailed comments and constructive suggestions. We hope our point-to-point responses can address your concerns. (Sorry for splitting the responses into multiple parts due to the limits on the character number.)
>
> >Q1. Not clear in experiments what LLMatch metric is used currently.
>
> Thanks for the comment.  We are a little unsure if you are asking about the specific *matching loss function* we are using in our experiments, or if you're asking about the *evaluation metrics* used.
>
> If your question is the former (re: matching loss function), we'd like to clarify that we are using Eq.7 (Section 3, under "Practical Implementation of LLP").  "Methods" section under Section 4.1 also clarifies: "LLP refers to MLP trained with our proposed relational KD (Equation (7))." Table 4 shows details about the performance of LLP when ablating parts of the loss function.
>
> If your question is the latter (re: evaluation metrics), we illustrate our evaluation metrics in "Evaluation Settings and Metrics" under Section 4.1: “For OGB datasets, we use their official metric (Hits@50 for OGB-Collab and Mean reciprocal Rank (MRR) for OGB-Citation2) following the public leaderboard. For other datasets, following previous works [1,2,3], we use Hits@20 as the main metric, which is also one of the main metrics on OGB datasets. We also report AUC performance in Appendix D.”  **To further improve the clarity, we added the evaluation metric information in the title of each experimental table.**
>
> [1] Yun et al., Neo-gnns: Neighborhood overlap-aware graph neural networks for link prediction. NeurIPS 2021.\
> [2] Zhang et al., Labeling trick: A theory of using graph neural networks for multi-node representation learning. NeurIPS 2021.\
> [3] Zhao et al., Learning from counterfactual links for link prediction. ICML 2022
>
> >Q2. Uses logit based matching based on a very old paper. And also doesn't showcase the motivation of proposal of logit based matching. It should be clearly substituted with recent paper evidence.
>
> Thanks for your suggestion. Although logit-based matching knowledge distillation (KD) [4] was originally proposed some time ago, it is still one of the most widely used KD methods in various tasks [5,6,11]. Several works theoretically analyzed its effectiveness: Phuong et al. [7] theoretically supported the success of logit-based matching KD by proving a faster convergence rate from it. Ji et al. [8] proved logit-based matching based KD can be beneficial by measuring the training difficulty caused by KD. Moreover, it also had been proved to be effective for knowledge transfer on graph data [9,10,11] in recent years.
>
> Besides its effectiveness, logit-based matching is a straightforward strategy for KD, where it directly aims to teach the student to generalize as the teacher does on the downstream task. It is easy to apply it on different downstream tasks. For example, Zhang et al. [11] utilized the soft logits generated by the teacher GNNs to help supervise the student MLP and achieved strong performance on node classification tasks, as recently as ICLR'22. Following the same strategy, we propose to use this straightforward and effective way for our tasks. **We have added this discussion about our motivation in Section 3.1 to make the motivation for its use clear.**
>
> [4] Hinton et al., Distilling the knowledge in a neural network. CVPR 2015. \
> [5] Furlanello et al., Born again neural networks. ICML 2018. \
> [6] Yang et al., Model compression with two-stage multi-teacher knowledge distillation for web question answering system. WSDM 2020.\
> [7] Phuong et al., Towards understanding knowledge distillation. ICML 2019.\
> [8] Ji et al., Knowledge distillation in wide neural networks: Risk bound, data efficiency and imperfect teacher. NeurIPS 2020.\
> [9] Yan et al., Tinygnn: Learning efficient graph neural networks. KDD 2020.\
> [10] Yang et al., Extract the knowledge of graph neural networks and go beyond it: An effective knowledge distillation framework. WWW 2021.\
> [11] Zhang et al., Graph-less neural networks: Teaching old mlps new tricks via distillation. ICLR 2022.

---

> ### Author Response · Authors · 2022-12-07
> **Looking forward to your response**
>
> Dear Reviewer Wrjf,
>
> We sincerely appreciate your valuable comments. We took significant efforts during the rebuttal to clarify misunderstandings and address your concerns with our responses.
>
> Since the discussion period is close to the end, could you kindly look at our responses and raise your score if you believe we have satisfactorily addressed your concerns? If you have any further concerns, we will be happy to respond.
>
> Best Regards,\
> LLP Authors

---

> ### Author Response · Authors · 2022-12-12
> **Last day reminder**
>
> Dear Reviewer Wrjf,
>
> Thank you again for your review! Today is the last day for the discussion. We do hope you can take a look at our responses to your comments and give us some feedback. We believe we have addressed all your questions and concerns in this revision. If so, please consider raising your score. If there are any further concerns, please let us know, and we will gladly address them.
>
> Thanks,\
> LLP Authors

---

### Author Response · Authors · 2022-11-29
**Reminder to Reviewers**

Dear Reviewers,

Thank you again for your constructive comments and suggestions.

As the end of the discussion is approaching, we would like to ask whether our response has addressed your concerns. If there are any further questions, please let us know, and we will be happy to respond.

Thanks again for your time. And if our response has addressed your concerns, we hope you will consider raising your score in light of the revised version and initial response.

Kind Regards,\
LLP Authors

---

### Decision · Program_Chairs · 2023-01-20

**Decision:**

Reject

**Justification For Why Not Higher Score:**

I carefully read the paper and the reviews/responses. The reasons are listed in the meta-review.

**Justification For Why Not Lower Score:**

N/A

**Metareview: Summary, Strengths And Weaknesses:**

This paper studies the knowledge distillation for graph link prediction. It proposes two novel loss functions based on ranking and distribution matching to distill knowledge from GNNs to MLPs. The motivation of using MLPs instead of GNNs is clear: in the production setting, GNNs cause severe delays due to neighbor explosion and complex computations. I agree with this motivation and am happy to see such a paper studying this problem. However, the following reasons prevent it from being accepted in the current form: 1) The performance drop from GNNs to the proposed LLP is too large on OGB datasets (especially on ogb-citation2, where the MRR drops from 82 to 53). This raises the concern of whether we still want to trust an MLP distilled from GNN, even it is much faster. I understand that on small datasets LLP even outperforms GNNs, but results on these datasets are far less convincing than those on OGB. 2) I have a strong concern on whether an MLP can really capture the important structural information learned by GNNs based only on the node features input. No matter how you modify the loss functions to match the outputs of GNN and MLP, the MLP does not take any structure as input---then it is impossible for MLP to capture similar structural features to those learned by a GNN, such as number of overlapped neighbors or distance between two nodes. This is a fundamental constraint of MLP. A possible solution is to let MLP additionally take some 1-hop neighbor statistics as input. 3) The technical contribution is limited. Only two heuristic matching losses are proposed without in-depth analysis. 4) Even before distillation, the GNN performance is far from those leading methods in the OGB leaderboard. This even worsens the performance after distillation.

Overall, I like the studied problem but the above shortcomings prevent me from accepting the paper now. I encourage the authors to adequately address the above problems in their revision, which will make the work more solid and convincing.